# Adapting LLMs to Time Series Forecasting via Temporal Heterogeneity Modeling and Semantic Alignment

## Abstract

Large Language Models (LLMs) have recently demonstrated impressive performance in natural language processing due to their strong generalization and sequence modeling capabilities. However, their direct application to time series forecasting remains challenging due to two fundamental issues: the inherent heterogeneity of temporal patterns and the modality gap between continuous numerical signals and discrete language representations. In this work, we propose **TALON** (Temporal-heterogeneity And Language-Oriented Network), a unified framework that enhances LLM-based forecasting by modeling temporal heterogeneity and enforcing semantic alignment. Specifically, we design a Heterogeneous Temporal Encoder that partitions multivariate time series into structurally coherent segments, enabling localized expert modeling across diverse temporal patterns. To bridge the modality gap, we introduce a Semantic Alignment Module that aligns temporal features with LLM-compatible representations, enabling effective integration of time series into language-based models while eliminating the need for handcrafted prompts during inference. Extensive experiments on seven real-world benchmarks demonstrate that TALON achieves superior performance across all datasets, with average MSE improvements of up to 11% over recent state-of-the-art methods, while maintaining higher efficiency. These results underscore the effectiveness of incorporating both pattern-aware and semantic-aware designs when adapting LLMs for time series forecasting. The code is available at: https://anonymous.4open.science/r/TALON-BB00.

## 1 Introduction

Time series forecasting plays a critical role in a wide range of real-world applications, spanning high-stakes domains such as healthcare monitoring (Jin et al., 2023) and power grid control (Shao et al., 2024), as well as everyday services including weather forecasting (Sun et al., 2021; Zhang et al., 2023; Price et al., 2025), traffic prediction (Jin et al., 2024c), and energy load estimation (Wu et al., 2024). To ensure reliable forecasting in such complex and dynamic environments, it is essential to effectively model long-range temporal dependencies (Nie et al., 2023; Liu et al., 2024c).

Recently, large language models (LLMs) have demonstrated remarkable generalization and representation capabilities across a wide range of language and vision tasks (Touvron et al., 2023; Liu et al., 2023; Achiam et al., 2023; Team, 2024; Liu et al., 2024b). Inspired by the shared sequential nature of time series and language data, recent research has explored LLMs as general-purpose forecasters for time series applications (Ansari et al., 2024; Jin et al., 2024a; Liu et al., 2024d), aiming to leverage their strong sequence modeling capabilities.

However, as illustrated in Figure 1 (a), multivariate time series often exhibit intrinsic heterogeneity, where different segments and variables follow diverse and evolving patterns (Shao et al., 2024; Sun et al., 2024; Qiu et al., 2025; Liu et al., 2025c; Shi et al., 2025). In contrast, LLMs are pretrained on text corpora with globally consistent grammatical structures, which limits their ability to handle fragmented or nonstationary temporal inputs. Moreover, time series are continuous and real-valued, governed by strong temporal dependencies, whereas LLMs are inherently designed for discrete, symbolic sequences (Ansari et al., 2024). This discrepancy in both structure and modality poses significant challenges for directly applying LLMs to time series forecasting (Liu et al., 2025a).

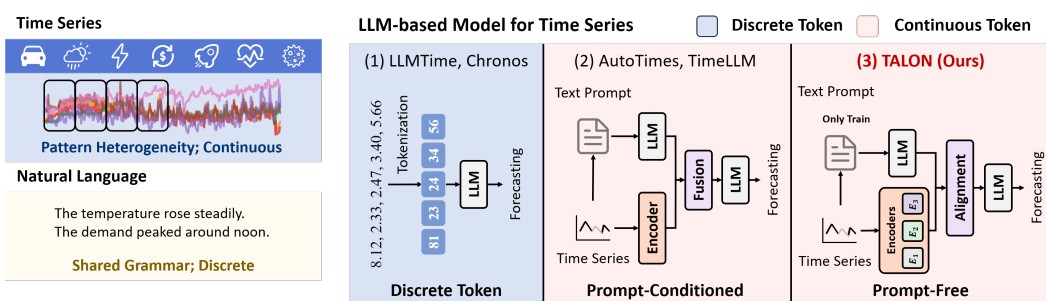

(a) Differences between time series and natural language.

(b) Comparison of LLM-based forecasting paradigms.

Figure 1: (a) Time series are continuous and structurally diverse, whereas natural language is discrete and syntactically uniform, posing a modality gap that hinders the direct application of LLMs to time series forecasting. (b) Our proposed TALON introduces a framework that integrates heterogeneous temporal encoding with contrastive semantic alignment, enabling pattern-aware and semantically grounded forecasting without relying on prompts during inference.

As shown in Figure 1 (b), existing LLM-based forecasting methods primarily fall into two categories: (1) Tokenization-based methods, which discretize continuous sequences into symbolic tokens (Gruver et al., 2023; Ansari et al., 2024); and (2) Prompt-conditioned methods, which prepend handcrafted textual templates to time series inputs (Liu et al., 2024d; Jin et al., 2024a). While both paradigms attempt to adapt LLMs to time series data, they fail to fully account for the modality gap. Specifically, they either disrupt temporal continuity, discard fine-grained numerical structure, or suffer from weak alignment and a reliance on handcrafted prompts.

To address these challenges, we propose **TALON** (Temporal-heterogeneity And Language-Oriented Network), a unified framework that bridges the modality gap by jointly modeling temporal heterogeneity and enforcing semantic alignment between time-series and language representations. First, we propose a **Heterogeneous Temporal Encoder (HTE)** to partition multivariate time series into structurally homogeneous segments based on their statistical and temporal properties, enabling pattern-aware expert modeling. Second, we introduce a **Semantic Alignment Module (SAM)** that aligns continuous features with LLM-compatible embeddings in a shared semantic space, eliminating the need for handcrafted prompts and bridging the modality gap. Finally, we employ a **LLM Forecasting Head (LFH)** that combines a pretrained LLM with lightweight projection layers to autoregressively generate future segments from the aligned representations. We evaluate TALON on seven real-world time series forecasting benchmarks, where it consistently outperforms both LLM-based and deep learning baselines across various prediction horizons. Our contributions are summarized as follows:

- We identify and characterize the modality misalignment problem in LLM-based time series forecasting from both structural and semantic perspectives, highlighting how the discrepancy between continuous signals and discrete language inputs limits existing paradigms.

- We propose TALON, a novel framework that integrates heterogeneous pattern decomposition and semantic alignment to enable fine-grained forecasting and cross-modal representation learning.

- Experimentally, TALON consistently outperforms state-of-the-art baselines across seven real-world forecasting benchmarks, achieving up to 11% reduction in MSE while improving both accuracy and generalization.

## 2 RELATED WORK

**Deep Learning for Time Series Forecasting.** Deep learning has become a cornerstone in time series forecasting, with various architectures designed to capture complex temporal dependencies. Convolutional neural networks are widely used to extract local temporal patterns and variable-wise dependencies (Wu et al., 2023; Eldele et al., 2024; Wang et al., 2025). More recently, Transformer-based models have gained popularity due to their global receptive fields and self-attention mechanisms, which enable long-range dependency modeling. For instance, PatchTST (Nie et al., 2023) proposes a channel-independent patching mechanism to decouple variable interactions,

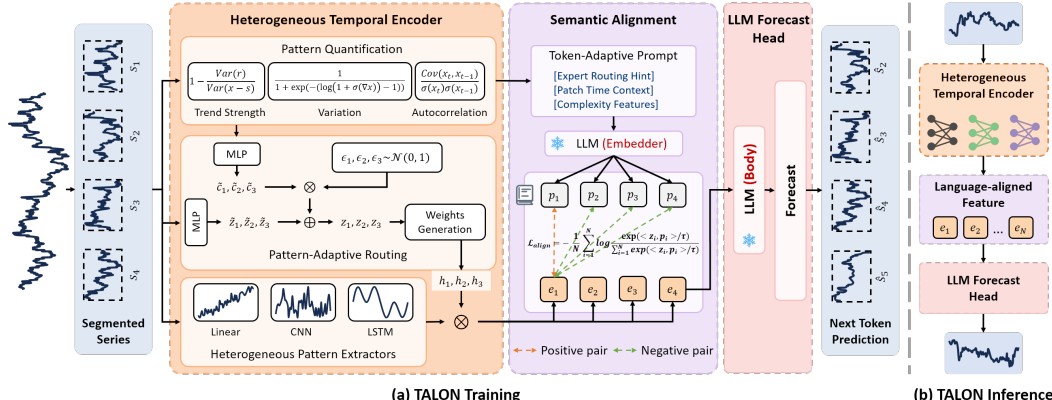

**(a) TALON Training** **(b) TALON Inference**

Figure 2: Overview of the TALON architecture. (1) Heterogeneous Temporal Encoder quantifies segment-level complexity and routes each segment to specialized experts via a pattern-based routing mechanism. (2) Semantic Alignment Module generates structured, token-level prompts that encode expert routing hints and temporal context, and applies contrastive learning to align time-series and language representations, thereby enabling semantic grounding without inference-time prompts. (3) LLM Forecasting Head takes the aligned features as input and performs autoregressive next-segment prediction. This design supports complexity-aware modeling, prompt-free inference, and semantically aligned forecasting under heterogeneous temporal patterns.

while iTransformer (Liu et al., 2024c) enhances multivariate modeling by treating each univariate series as an individual token. To further address the heterogeneity of temporal patterns, several methods introduce mechanisms such as mixture-of-experts (Ni et al., 2024; Qiu et al., 2025; Liu, 2025) and subspace-based pattern grouping (Sun et al., 2024), improving robustness to non-stationary and diverse dynamics. Despite these advances, most existing methods remain constrained by limited parameterization and small-scale training corpora (Chen et al., 2020; Liu et al., 2021; Cai et al., 2024; Liu et al., 2024e).

**Large Language Models for Time Series.** Motivated by the sequential nature shared between time series and language, such as local-to-global dependency structures and autoregressive generation, recent studies have explored adapting LLMs to time series forecasting (Gruver et al., 2023; Jin et al., 2024b). One line of work discretizes time series into symbolic tokens via quantization or pattern clustering, enabling direct utilization of token-based LLMs (Gruver et al., 2023; Ansari et al., 2024). Another line of research retains raw numerical inputs and leverages textual prompts to provide contextual guidance (Liu et al., 2024d; Jin et al., 2024a; Niu et al., 2025). While these approaches benefit from the generalization capabilities of pretrained LLMs, they typically overlook the pattern and semantic mismatch between natural language and continuous time series, leading to limited scalability and suboptimal representation alignment.

## 3 PRELIMINARIES

Given a multivariate input sequence $X = (x_{t-L+1}, \ldots, x_t) \in \mathbb{R}^{L \times C}$, the goal of time series forecasting is to predict the future values $Y = (x_{t+1}, \ldots, x_{t+H}) \in \mathbb{R}^{H \times C}$, where $L$ is the look-back window length, $H$ is the forecasting horizon, and $C$ is the number of variables. The task is to learn a predictive function $f_\theta$ such that $Y = f_\theta(X)$.

## 4 METHOD

### 4.1 OVERALL ARCHITECTURE

As illustrated in Figure 2, our proposed framework **TALON** consists of three key components: the Heterogeneous Temporal Encoder (HTE), the Semantic Alignment Module (SAM), and the LLM Forecasting Head (LFH).

To focus on modeling temporal variations, we follow the channel-independent strategy (Liu et al., 2024d), decomposing the multivariate input into $C$ separate univariate sequences. Each univariate sequence is further segmented into $N$ consecutive non-overlapping patches of length $S$, with each patch denoted as $s_i = \{x_{(i-1)S+1}, \ldots, x_{iS}\} \in \mathbb{R}^S, i = 1, \cdots, N$.

The HTE module extracts token-level statistical features from each patch and dynamically routes it to a specialized expert (e.g., Linear, CNN, LSTM) via a learnable gating mechanism, enabling localized and pattern-aware temporal modeling. Next, the SAM constructs token-adaptive prompts based on the patch's complexity and temporal context. These prompts are processed by a frozen LLM to produce semantic embeddings in the language modality. To bridge the modality gap between continuous time-series features and discrete language representations, we introduce a fine-grained contrastive alignment loss at the token level. This encourages the time-series-derived representations to align closely with the language embeddings, effectively transforming them into language-aligned features suitable for LLM-based forecasting. Finally, the LFH takes the aligned embeddings as input and employs a autoregressive decoder, consisting of a frozen LLM and a linear projection layer, to generate forecasting outputs. This design supports variable prediction lengths while maintaining low inference cost. We elaborate on each module in the following subsections.

## 4.2 HETEROGENEOUS TEMPORAL ENCODER

Multivariate time series often exhibit complex and heterogeneous temporal dynamics, including diverse trends, fluctuations, and long-range dependencies across variables and time (Shao et al., 2024). To effectively model such variability, we propose the Heterogeneous Temporal Encoder (HTE), which learns pattern-aware representations by dynamically adapting expert selection to the complexity and temporal structure of each input patch.

As shown in Figure 2, HTE consists of three key components: (1) Pattern Quantification, (2) Pattern-Adaptive Routing, and (3) Heterogeneous Pattern Extractors.

**Pattern Quantification.** To characterize the local temporal structure of each patch, HTE computes a compact set of interpretable token-level statistical features (e.g., trend strength, variation, and autocorrelation). These features quantify local temporal dynamics and serve as the basis for routing each patch to a specialized modeling branch tailored to distinct temporal behaviors.

Given a univariate patch $s_i \in \mathbb{R}^S$, we compute three descriptors: trend strength ($c_1$), local variation ($c_2$), and autocorrelation coefficient ($c_3$) (Qiu et al., 2024; Li et al., 2024). These features form a quantification vector $c_i = [c_1, c_2, c_3] \in \mathbb{R}^3$, which characterizes the local structure of $s_i$ and serves as the input to the expert routing mechanism. The specific calculation formulas are provided in the Appendix C.

**Pattern-Adaptive Routing.** Inspired by Variational Autoencoder-style stochastic modeling (Kingma et al., 2013), we introduce latent uncertainty into the expert selection process by encoding both the input patch $s_i$ and its complexity $c_i$ into latent scores. Specifically, we compute:

$$\tilde{z}_i = \text{ReLU}(s_i W_0^t) W_1^t, \tag{1}$$
$$\tilde{c}_i = \text{ReLU}(c_i W_0^c) W_1^c, \tag{2}$$

where $W_0^t \in \mathbb{R}^{S \times d}$, $W_0^c \in \mathbb{R}^{3 \times d}$, and $W_1^t, W_1^c \in \mathbb{R}^{d \times K}$ for $K$ experts.

We inject Gaussian noise $\epsilon_i \sim \mathcal{N}(0, 1)$ and compute routing logits:

$$z_i = \tilde{z}_i + \epsilon_i \cdot \text{Softplus}(\tilde{c}_i), \tag{3}$$
$$h_i = z_i W^H, \tag{4}$$

where $W^H \in \mathbb{R}^{K \times K}$ is a projection matrix that maps the latent vector to the expert scoring space, and $h, \epsilon \in \mathbb{R}^K$. To promote sparsity, we retain the top-$k$ entries in $h_i$ before applying softmax:

$$G(s_i) = \text{Softmax}(\text{KeepTopk}(h_i, k)), \tag{5}$$
$$\text{KeepTopk}(h_i, k)_j = \begin{cases} h_{i,j}, & \text{if } j \in \text{Topk}(h_i), \\ -\infty, & \text{otherwise.} \end{cases} \tag{6}$$

**Heterogeneous Pattern Extractors.** Unlike previous methods that adopt a unified architecture for all time segments (Nie et al., 2023; Liu et al., 2024c) or apply homogeneous experts uniformly across patches (Sun et al., 2024; Qiu et al., 2025), we recognize that time series often exhibit diverse temporal patterns, such as trends, local fluctuations, and long-range dependencies, which motivate a heterogeneous modeling strategy. To this end, we design a lightweight expert pool comprising three complementary branches that provide diverse temporal modeling capacities, enabling the framework

to adapt to heterogeneous temporal dynamics and improve robustness across varied forecasting scenarios:

1. Linear Expert for modeling trend-like patterns: $\mathbf{e}_i^{\text{Linear}} = s_i \cdot \mathbf{W}_{\text{Linear}}$.

2. CNN Expert for capturing local dependencies: $\mathbf{e}_i^{\text{CNN}} = \mathbf{W}_{\text{proj}} \cdot (\text{Conv}_2(\text{ReLU}(\text{Conv}_1(s_i))))$.

3. LSTM Expert for modeling long-term memory: $\mathbf{e}_i^{\text{LSTM}} = \mathbf{W}_{\text{proj}} \cdot \text{LSTM}(s_i)_{[-1]}$, where $\text{LSTM}(s_i)_{[-1]}$ denotes the hidden state of the last time step given input $s_i$.

Let $\mathbf{e}_i^j$ denote the output of the $j$-th expert. The final representation for patch $s_i$ is computed as a weighted aggregation over all expert outputs:

$$\mathbf{e}_i = \sum_{j=1}^{K} G(s_i)_j \cdot \mathbf{e}_i^j, \tag{7}$$

where $G(s_i) \in \mathbb{R}^K$ is the sparse gating vector produced by the pattern-adaptive routing mechanism.

**Expert Regularization.** To prevent expert collapse and promote diverse expert usage, we incorporate a load-balancing regularization term inspired by (Shazeer et al., 2017):

$$\mathcal{L}_{\text{MoE}} = \mathcal{L}_{\text{importance}} + \mathcal{L}_{\text{load}}. \tag{8}$$

Here, $\mathcal{L}_{\text{importance}}$ minimizes the coefficient of variation across expert gate importance scores, while $\mathcal{L}_{\text{load}}$ penalizes imbalanced token-to-expert assignments. This regularization stabilizes training and promotes more efficient utilization of the expert capacity.

### 4.3 SEMANTIC ALIGNMENT MODULE

Most existing LLM-based time series forecasting approaches rely on static, global prompts shared across all tokens (Jin et al., 2024a), which fail to capture the temporal heterogeneity inherent in multivariate time series and limit generalization to local patterns. Furthermore, these methods typically adopt shallow alignment strategies (Liu et al., 2024d), resulting in representations that are misaligned with the architecture of LLMs and fail to fully exploit their reasoning capabilities.

To address these limitations, we propose the Semantic Alignment Module (SAM), which performs fine-grained token-level alignment between temporal features and their corresponding textual semantics via contrastive learning. By generating token-adaptive prompts and embedding both modalities into a shared latent space, SAM enables the LLM to reason in a space that is both semantically meaningful and temporally aware.

**Token-Adaptive Prompt.** Unlike language tokens that follow consistent syntactic structures, time series tokens encode heterogeneous temporal semantics. Applying a uniform prompt across such tokens can obscure informative variations. Inspired by recent advances in visual prompting (Liu et al., 2025f), we extend the idea of differentiated prompts to time series.

We construct token-adaptive prompts using interpretable statistical descriptors for each token, with each prompt integrating three aspects: (1) expert routing hints, (2) patch-wise temporal context, and (3) complexity-aware features. Motivated by the attention analysis in (Liu et al., 2025a), we place numerical features at the end of the prompt to guide the LLM's focus toward informative value tokens. These elements are tokenized using the LLM tokenizer to yield prompt embeddings $p_i$.

> **Token-Adaptive Prompt**
>
> [Expert Routing Hint]
> The available expert types are: Linear, CNN, LSTM.
>
> [Patch Time Context]
> This patch consists of $\langle \text{token\_len} \rangle$ time steps, from $\langle \text{patch\_start} \rangle$ to $\langle \text{patch\_end} \rangle$.
> It is part of a longer input window, which spans from $\langle \text{x\_start} \rangle$ to $\langle \text{x\_end} \rangle$ and contains $\langle \text{seq\_len} \rangle$ time steps.
>
> [Complexity Features]
> Trend Strength: $\langle c_1 \rangle$.
> Local Variation: $\langle c_2 \rangle$.
> Temporal Dependency: $\langle c_3 \rangle$.

**Semantic Alignment.** To bridge the modality gap between temporal signals and language representations, we design a contrastive alignment mechanism that injects prompt semantics into temporal

features at the token level. For each token $i$, we align its temporal feature $e_i$ with its associated prompt embedding $p_i$ via a contrastive objective:

$$\mathcal{L}_{\text{align}} = -\frac{1}{N}\sum_{i=1}^{N} log \frac{exp(\langle e_i, p_i \rangle / \tau)}{\sum_{i=1}^{N} exp(\langle e_i, p_i \rangle / \tau)}, \quad (9)$$

where $\langle \cdot, \cdot \rangle$ denotes cosine similarity, $\tau$ is a temperature parameter, and all vectors are $\ell_2$-normalized.

This alignment enforces temporal features to reside in a shared semantic space with their corresponding prompts, thereby enabling the LLM to interpret temporal patterns with enhanced semantic consistency.

## 4.4 LLM FORECASTING HEAD

By aligning temporal features with language semantics, we enable the LLM to operate on time series in a semantically grounded representation space. The aligned features $e_i$ are first passed through a frozen pretrained LLM for deep contextual reasoning, after which a lightweight decoder projects the resulting representations into future predictions:

$$\hat{Y} = \text{MLP}(\text{LLM}(e)). \quad (10)$$

Our autoregressive decoding allows flexible forecasting without retraining for different horizons, fully utilizing LLMs' inherent capacity for multi-step generation (Liu et al., 2024d).

The final training objective jointly optimizes forecasting accuracy, expert utilization, and semantic alignment:

$$\mathcal{L} = \mathcal{L}_{\text{MSE}} + \alpha\mathcal{L}_{\text{MoE}} + \beta\mathcal{L}_{\text{align}}. \quad (11)$$

This formulation enables accurate and generalizable forecasts while maintaining an efficient decoding pipeline.

## 4.5 INFERENCE PIPELINE

As shown in Figure 2, the inference process is streamlined and fully prompt-free. The input time series is first segmented into patches $s_1, s_2, \ldots, s_N$, and each patch is processed by the Heterogeneous Temporal Encoder. A gating mechanism aggregates outputs from the top-$k$ experts, yielding semantically enriched features $e_1, e_2, \ldots, e_N$, which are then passed through a frozen pretrained LLM to perform autoregressive forecasting.

By eliminating the need for textual prompts and semantic alignment during inference, our framework supports efficient, pattern-aware forecasting with minimal computational overhead. This design enables faster inference and enhanced deployment flexibility, while retaining the representational benefits of heterogeneous expert modeling.

# 5 EXPERIMENT

## 5.1 DATA AND EXPERIMENT SETTING

**Dataset.** We evaluate the long-term forecasting performance across seven widely-used time series benchmarks, including ETT datasets (ETTh1, ETTh2, ETTm1, ETTm2), Weather, Electricity, and Traffic. These datasets are standard benchmarks in the long-term forecasting literature (Liu et al., 2024d). Detailed descriptions are provided in Appendix A.1.

**Baselines and Evaluation.** We compare TALON against state-of-the-art baselines from two categories: (1) LLM-based forecasting methods, including LangTime (Niu et al., 2025), CALF (Liu et al., 2025b), AutoTimes (Liu et al., 2024d), TimeLLM (Jin et al., 2024a), and FPT (Zhou et al., 2023); (2) Deep learning-based forecasting models, including SimpleTM (Chen et al., 2025), Timer_XL (Liu et al., 2025e), TimeMixer (Wang et al., 2024), iTransformer (Liu et al., 2024c), PatchTST (Nie et al., 2023), and TimesNet (Wu et al., 2023).

**Implementation Details.** Following the common setup in (Liu et al., 2024d), we fix the input lookback window size to $L = 672$ for all experiments and use pre-trained GPT2 based model (Radford et al., 2019) with the first 6 Transformer layers as our backbone. To ensure fair comparisons, we rerun all baselines. All the experiments are conducted using PyTorch (Paszke et al., 2019) on NVIDIA A100 GPUs.

Table 1: Multivariate forecasting (672-pred-{96, 192, 336, 720}) results under the one-for-all setting. Following Liu et al. (2024d), a single model is trained on a 96-step prediction horizon and evaluated on all horizons using rolling forecasting. The best results are in **bold**, and the second-best are *underlined*. Averaged results are reported here and full results are provided in Table 8. IMP denotes the average MSE and MAE reduction of TALON over each baseline across seven datasets.

| Model | LLM-based methods | | | | | | | | | | | | Deep learning forecasting methods | | | | | | | | | | | |
|---|---|---|---|---|---|---|---|---|---|---|---|---|---|---|---|---|---|---|---|---|---|---|---|---|
| | TALON (Ours) | | LangTime (2025) | | CALF (2025b) | | AutoTimes (2024d) | | TimeLLM (2024a) | | FPT (2023) | | SimpleTM (2025) | | Timer_XL (2025e) | | TimeMixer (2024) | | iTransformer (2024c) | | PatchTST (2023) | | TimesNet (2023) | |
| Metric | MSE | MAE | MSE | MAE | MSE | MAE | MSE | MAE | MSE | MAE | MSE | MAE | MSE | MAE | MSE | MAE | MSE | MAE | MSE | MAE | MSE | MAE | MSE | MAE |
| ETTh1 | **0.386** | **0.420** | 0.406 | 0.422 | 0.416 | 0.429 | 0.402 | 0.428 | 0.542 | 0.520 | 0.422 | 0.437 | 0.424 | 0.450 | 0.407 | 0.429 | 0.418 | 0.434 | 0.432 | 0.451 | 0.441 | 0.451 | 0.495 | 0.489 |
| ETTh2 | **0.355** | **0.395** | 0.364 | 0.399 | 0.373 | 0.419 | 0.400 | 0.431 | 0.416 | 0.446 | 0.370 | 0.407 | 0.367 | 0.414 | 0.377 | 0.414 | 0.385 | 0.417 | 0.399 | 0.423 | 0.392 | 0.429 | 0.455 | 0.463 |
| ETTm1 | **0.345** | **0.380** | 0.398 | 0.405 | 0.367 | 0.417 | 0.364 | 0.389 | 0.477 | 0.463 | 0.365 | 0.401 | 0.358 | 0.386 | 0.371 | 0.392 | 0.411 | 0.409 | 0.377 | 0.405 | 0.360 | 0.392 | 0.505 | 0.442 |
| ETTm2 | **0.259** | **0.319** | 0.262 | 0.323 | 0.281 | 0.341 | 0.277 | 0.327 | 0.310 | 0.359 | 0.283 | 0.337 | 0.268 | 0.325 | 0.281 | 0.333 | 0.277 | 0.330 | 0.282 | 0.338 | 0.284 | 0.341 | 0.293 | 0.347 |
| Weather | **0.239** | **0.278** | 0.265 | 0.282 | 0.255 | 0.298 | 0.252 | 0.290 | 0.271 | 0.308 | 0.248 | 0.284 | 0.247 | 0.282 | 0.322 | 0.355 | 0.244 | 0.282 | 0.258 | 0.286 | 0.247 | 0.284 | 0.260 | 0.291 |
| ECL | **0.162** | **0.255** | 0.178 | 0.272 | 0.239 | 0.296 | 0.168 | 0.261 | 0.185 | 0.288 | 0.257 | 0.354 | 0.167 | 0.261 | 0.173 | 0.272 | 0.167 | 0.257 | 0.167 | 0.260 | 0.180 | 0.283 | 0.207 | 0.304 |
| Traffic | **0.373** | **0.253** | 0.418 | 0.273 | 0.891 | 0.442 | 0.379 | 0.265 | 0.414 | 0.305 | 0.428 | 0.312 | 0.436 | 0.317 | 0.378 | 0.256 | 0.442 | 0.321 | 0.384 | 0.272 | 0.408 | 0.298 | 0.619 | 0.330 |
| IMP. | -- | -- | 7% | 10% | 17% | 21% | 5% | 11% | 17% | 20% | 11% | 16% | 6% | 14% | 9% | 12% | 8% | 14% | 7% | 12% | 8% | 14% | **22%** | **20%** |

Table 2: Multivariate forecasting (672-pred-{96, 192, 336, 720}) results under the one-for-one setting. A separate model is trained and evaluated for each prediction horizon. The best results are in **bold**, and the second-best are *underlined*. Averaged results are reported here and full results are provided in Table 9.

| Models | One-for-all | | Trained respectively on specific lookback / prediction length | | | | | | | | | | | | | | | | | | | | | |
|---|---|---|---|---|---|---|---|---|---|---|---|---|---|---|---|---|---|---|---|---|---|---|---|---|
| | TALON (Ours) | | LangTime (2025) | | CALF (2025b) | | AutoTimes (2024d) | | TimeLLM (2024a) | | FPT (2023) | | SimpleTM (2025) | | Timer_XL (2025e) | | TimeMixer (2024) | | iTransformer (2024c) | | PatchTST (2023) | | TimesNet (2023) | |
| Metric | MSE | MAE | MSE | MAE | MSE | MAE | MSE | MAE | MSE | MAE | MSE | MAE | MSE | MAE | MSE | MAE | MSE | MAE | MSE | MAE | MSE | MAE | MSE | MAE |
| ETTh1 | **0.386** | **0.420** | 0.451 | 0.447 | 0.440 | 0.452 | 0.457 | 0.466 | 0.578 | 0.529 | 0.438 | 0.446 | 0.422 | 0.449 | 0.450 | 0.455 | 0.428 | 0.442 | 0.451 | 0.465 | 0.468 | 0.467 | 0.484 | 0.489 |
| ETTh2 | **0.355** | **0.395** | 0.388 | 0.408 | 0.366 | 0.402 | 0.390 | 0.421 | 0.435 | 0.455 | 0.396 | 0.427 | 0.361 | 0.395 | 0.370 | 0.407 | 0.374 | 0.409 | 0.400 | 0.426 | 0.417 | 0.438 | 0.433 | 0.465 |
| ETTm1 | **0.345** | **0.380** | 0.415 | 0.414 | 0.363 | 0.393 | 0.411 | 0.418 | 0.406 | 0.417 | 0.359 | 0.390 | 0.356 | 0.390 | 0.359 | 0.391 | 0.418 | 0.423 | 0.372 | 0.403 | 0.387 | 0.409 | 0.444 | 0.434 |
| ETTm2 | **0.259** | **0.319** | 0.266 | 0.323 | 0.266 | 0.321 | 0.307 | 0.353 | 0.290 | 0.345 | 0.274 | 0.330 | 0.269 | 0.329 | 0.276 | 0.329 | 0.269 | 0.327 | 0.274 | 0.335 | 0.289 | 0.343 | 0.303 | 0.353 |
| Weather | **0.239** | **0.278** | 0.277 | 0.294 | 0.241 | 0.281 | 0.249 | 0.287 | 0.273 | 0.313 | 0.242 | 0.282 | 0.244 | 0.281 | 0.324 | 0.356 | 0.262 | 0.293 | 0.261 | 0.290 | 0.240 | 0.280 | 0.252 | 0.290 |
| ECL | **0.162** | **0.255** | 0.174 | 0.268 | 0.165 | 0.262 | 0.172 | 0.269 | 0.176 | 0.276 | 0.166 | 0.263 | 0.166 | 0.261 | 0.170 | 0.258 | 0.166 | 0.257 | 0.163 | 0.258 | 0.166 | 0.267 | 0.203 | 0.307 |
| Traffic | **0.373** | **0.253** | 0.469 | 0.378 | 0.386 | 0.265 | 0.385 | 0.268 | 0.402 | 0.284 | 0.408 | 0.288 | 0.453 | 0.331 | 0.382 | 0.263 | 0.404 | 0.285 | 0.386 | 0.275 | 0.397 | 0.279 | 0.622 | 0.329 |
| IMP. | -- | -- | 12% | 18% | 4% | 10% | 10% | 13% | 15% | 18% | 6% | 12% | 6% | 14% | 9% | 13% | 8% | 13% | 7% | 13% | 9% | 14% | **20%** | **20%** |

## 5.2 TIME SERIES FORECASTING

**Setups.** We consider two evaluation protocols to assess the forecasting performance of our model: (1) To evaluate the generalization capability of one-for-all forecasting, we adopt the rolling forecast setting (Liu et al., 2024d; 2025e), where a single model is trained on a 96-step prediction horizon and then directly applied to all other horizons. During inference, the predicted values are recursively fed into the lookback window to generate subsequent predictions. (2) For the conventional one-for-one setting, we follow the standard multivariate evaluation protocol adopted by TimesNet (Wu et al., 2023), where a separate model is trained and evaluated for each prediction horizon.

**Results.** The average forecasting results are reported in Table 1 and Table 2. In the one-for-all setting (Table 1), TALON consistently achieves the lowest MSE across all seven datasets, with an average improvement of up to 10% over state-of-the-art deep forecasters and 12% over recent LLM-based methods. In the conventional one-for-one setting (Table 2), it further achieves state-of-the-art performance with up to 20% MSE reduction. These results highlight TALON's strong generalization capability and its effectiveness in modeling heterogeneous and evolving temporal patterns.

## 5.3 ZERO-SHOT FORECASTING

**Setups.** LLMs have exhibited remarkable zero-shot generalization capabilities across various domains (Brown et al., 2020). To assess whether TALON inherits this ability, we adopt the widely used zero-shot forecasting protocol (Jin et al., 2024a; Liu et al., 2024d), where a model is trained on a source domain and directly evaluated on an unseen target domain without any fine-tuning. Following this setting, we use the ETT benchmark family and conduct evaluations across multiple cross-domain scenarios, including both resolution shifts and domain shifts among ETT variants. As in the full-shot experiments, we adopt the long-term forecasting protocol for evaluation.

**Results.** The zero-shot forecasting results are summarized in Table 3. TALON consistently achieves the best MSE performance in 4 tasks, outperforming all compared methods. Specifically, it achieves 8%~20% relative MSE improvement, demonstrating robust generalization across diverse transfer scenarios, including both resolution-level shifts and cross-domain adaptations. These results validate the effectiveness of TALON in capturing local temporal structures and leveraging LLM-based semantic alignment for strong transferability. Full results are provided in Table 10.

Table 3: Zero-shot forecasting result.

| Models | TALON | | LangTime | | AutoTimes | | Timer_XL | |
|---|---|---|---|---|---|---|---|---|
| | (Ours) | | (2025) | | (2024d) | | (2025e) | |
| Metric | MSE | MAE | MSE | MAE | MSE | MAE | MSE | MAE |
| h1→h2/m1/m2 | **0.478** | **0.446** | 0.622 | 0.490 | 0.506 | 0.451 | 0.512 | 0.461 |
| h2→h1/m1/m2 | **0.554** | **0.493** | 0.753 | 0.545 | 0.712 | 0.547 | 0.592 | 0.514 |
| m1→h1/h2/m2 | **0.432** | 0.434 | 0.474 | 0.451 | 0.436 | **0.433** | 0.480 | 0.458 |
| m2→h1/h2/m1 | **0.457** | **0.456** | 0.588 | 0.516 | 0.519 | 0.479 | 0.494 | 0.470 |
| IMP. | – – | – – | 20% | 8% | 10% | 4% | 8% | 4% |

Table 4: Comparison with MoE-based methods.

| Models | TALON | | FreqMoE | | MoFE-time | | TimeMoE | | TFPS | |
|---|---|---|---|---|---|---|---|---|---|---|
| | (Ours) | | (2025) | | (2025d) | | (2025) | | (2024) | |
| Metric | MSE | MAE | MSE | MAE | MSE | MAE | MSE | MAE | MSE | MAE |
| ETTh1 | **0.386** | **0.420** | 0.440 | 0.429 | 0.396 | 0.423 | 0.402 | 0.429 | 0.448 | 0.443 |
| ETTh2 | **0.355** | **0.395** | 0.367 | 0.396 | 0.438 | 0.439 | 0.472 | 0.458 | 0.380 | 0.403 |
| ETTm1 | **0.345** | **0.380** | 0.375 | 0.396 | 0.391 | 0.420 | 0.407 | 0.427 | 0.395 | 0.407 |
| ETTm2 | **0.259** | **0.319** | 0.271 | 0.338 | 0.278 | 0.347 | 0.324 | 0.377 | 0.276 | 0.321 |
| IMP. | – – | – – | 7% | 3% | 10% | 7% | **16%** | **11%** | 10% | 4% |

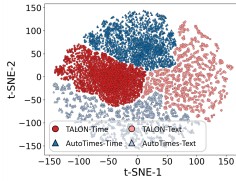
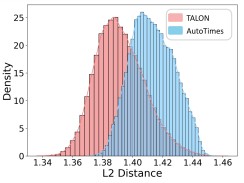

(a) t-SNE        (b) L2 distribution

Figure 3: Visualization of time–text alignment for TALON and AutoTimes on ETTh1-96.

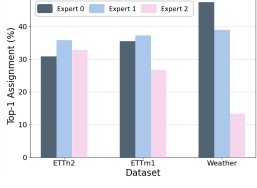

Figure 4: Analysis of expert assignment distributions.

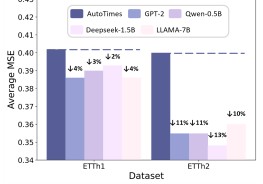

Figure 5: TALON generalization with different LLM backbones.

## 5.4 COMPARED WITH MoE-BASED METHODS

We compare TALON with recent MoE-based forecasting approaches. As shown in Table 4 (with full results reported in Table 11), TALON achieves the best MSE scores across all datasets, with an average improvement of 7% to 16% over existing methods. These consistent gains highlight the advantage of heterogeneous expert modeling: by incorporating diverse inductive biases, TALON adapts to evolving temporal dynamics and heterogeneous patterns across segments, leading to more reliable and robust forecasts. This demonstrates the importance of architectural diversity in enhancing model generalization and handling non-stationary dynamics across time series segments.

## 5.5 MODEL ANALYSIS

**Cross-Modal Embedding Alignment Analysis.** To evaluate the quality of cross-modal alignment, we analyze both the spatial structure and the quantitative similarity between time-series and textual embeddings. As shown in Figure 3 (a), the t-SNE visualization shows that TALON's temporal and textual embeddings form more compact clusters, indicating stronger semantic coupling. In contrast, AutoTimes exhibits a more scattered distribution, suggesting weaker alignment between modalities. We also compute the L2 distance between aligned time-text embedding pairs across the test set. As shown in Figure 3 (b), TALON achieves a significantly smaller mean distance than AutoTimes, confirming its stronger cross-modal correspondence.

**Expert Assign.** Figure 4 illustrates the expert assignment distributions of TALON across ETTh2, ETTm1, and Weather. Each bar indicates the percentage of input segments that are most confidently routed to a given expert. We observe that the expert utilization patterns vary significantly across datasets. For example, the Weather dataset shows a strong preference for Expert 0, whereas ETTh2 and ETTm1 exhibit more balanced and diverse assignments, indicating greater temporal complexity and higher pattern heterogeneity (Sun et al., 2024). This variation highlights TALON's ability to adaptively route segments to specialized experts based on underlying pattern characteristics, validating the effectiveness of its pattern-aware routing mechanism.

**Generality.** Previous LLM4TS approaches (Zhou et al., 2023; Jin et al., 2024a) typically target specific language models. In contrast, TALON is designed to be compatible with any decoder-only LLM. We evaluate this generality by replacing the default GPT-2 backbone with representative alternatives: Qwen (Team, 2024), Deepseek (Liu et al., 2024a), and LLaMA (Touvron et al., 2023). We choose AutoTimes as the baseline, as it exhibits the smallest relative performance improvement (5% in MSE) under TALON in Table 1. As shown in Figure 5, TALON consistently outperforms AutoTimes across all datasets and LLMs, with relative MSE reductions annotated on each bar. These results confirm that our framework is reliably enhances forecasting performance regardless of the underlying LLM. Full results are provided in Table 12.

Table 5: Performance of ablation studies.

| Models | ETTh1 | | ETTh2 | | ETTm1 | | ETTm2 | |
|---|---|---|---|---|---|---|---|---|
| Metric | MSE | MAE | MSE | MAE | MSE | MAE | MSE | MAE |
| TALON | **0.386** | **0.420** | **0.355** | **0.395** | **0.345** | **0.380** | **0.259** | **0.319** |
| w/o HTE | 0.403 | 0.427 | 0.365 | 0.405 | 0.347 | 0.380 | 0.267 | 0.325 |
| w/o HTE_R | 0.403 | 0.426 | 0.360 | 0.400 | 0.349 | 0.382 | 0.268 | 0.323 |
| w/o SAM | 0.393 | 0.422 | 0.367 | 0.408 | 0.350 | 0.382 | 0.266 | 0.319 |
| w/o Prompt | 0.389 | 0.419 | 0.363 | 0.406 | 0.352 | 0.383 | 0.266 | 0.322 |
| w/o LLM | 0.418 | 0.435 | 0.386 | 0.434 | 0.396 | 0.411 | 0.280 | 0.333 |

Table 6: Effectiveness of heterogeneous experts in HTE.

| Heterogeneous Experts | | | ETTh1 | | ETTh2 | | ETTm1 | | ETTm2 | |
|---|---|---|---|---|---|---|---|---|---|---|
| Linear | CNN | LSTM | MSE | MAE | MSE | MAE | MSE | MAE | MSE | MAE |
| ✓ | ✓ | ✓ | **0.386** | **0.420** | **0.355** | **0.395** | **0.345** | **0.380** | **0.259** | **0.319** |
| ✗ | ✓ | ✓ | 0.389 | 0.419 | 0.370 | 0.407 | 0.354 | 0.385 | 0.266 | 0.322 |
| ✓ | ✗ | ✓ | 0.401 | 0.426 | 0.360 | 0.400 | 0.353 | 0.386 | 0.265 | 0.320 |
| ✓ | ✓ | ✗ | 0.393 | 0.422 | 0.363 | 0.405 | 0.350 | 0.382 | 0.263 | 0.320 |

**Ablation Studies.** We conduct ablation studies to evaluate the contributions of TALON's key components. As shown in Table 5, removing the full HTE module (w/o HTE) increases average MSE by 8.6%, while disabling only the routing mechanism (w/o HTE_R) leads to a 8.1% increase, highlighting the value of expert specialization and routing. Disabling SAM (w/o SAM) results in a 7.2% increase in MSE, demonstrating its benefit in aligning temporal and textual representations. Replacing our token-adaptive prompt with a static TimeLLM-style prompt (w/o Prompt) leads to 5.6% degradation, validating the design of context-aware prompt construction. Removing the LLM (w/o LLM) causes the most significant drop, with a 33.9% increase in MSE, indicating the essential role of LLM's reasoning capacity. These results confirm that each module meaningfully contributes to TALON's performance, and their combination produces a synergistic effect for modeling complex, heterogeneous temporal dynamics.

**Analysis of HTE.** Table 6 validates the effectiveness of the HTE design. The fully heterogeneous setup consistently achieves the best performance across all datasets. In contrast, removing any single expert type leads to notable performance degradation (8.1%, 7.9%, and 5.8%, respectively). These results underscore the complementary nature of distinct temporal modeling perspectives. Their integration enables the model to adapt to diverse temporal patterns within multivariate time series, thereby enhancing generalization across different forecasting scenarios.

**Efficiency Analysis.** As shown in Figure 6, we compare TALON's efficiency with other LLM-based models on ETTh1-96. TALON achieves the lowest MSE while maintaining a compact model size ($\sim$ 1.7M) and fast inference ($\sim$ 2s), showing that careful architectural design can improve accuracy without increasing computational cost. This efficiency stems from TALON's lightweight temporal encoder and prompt-free semantic alignment, which together reduce input redundancy by removing handcrafted prompts and mitigate input complexity by preserving the temporal continuity and numerical precision of the original series.

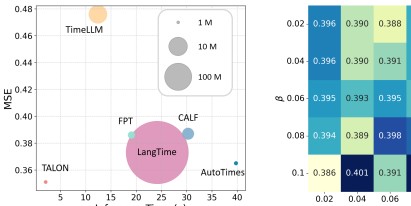

Figure 6: Efficiency comparison across LLM-based forecasters. Figure 7: Parameter sensitivity of $\alpha$ and $\beta$ on the ETTh1 dataset.

**Parameter Sensitivity.** We assess the robustness of our method to $\alpha$ and $\beta$ via a grid search, reporting MSE results on ETTh1 in Figure 7 and on other datasets in Figure 8. The performance remains relatively stable across a wide range of $\alpha$ and $\beta$ values, demonstrating that our model is not overly sensitive to specific hyperparameter settings and can deliver robust performance without extensive hyperparameter tuning. We provide additional analysis on the effect of top-$k$ expert selection in the Appendix D.5, and observe that activating multiple experts better captures pattern heterogeneity and improves forecasting performance.

# 6 CONCLUSION

This paper presents TALON, a novel framework for time series forecasting that integrates temporal heterogeneity modeling and semantic alignment within a unified foundation model architecture. By incorporating a heterogeneous temporal encoder and a semantic-aware fusion mechanism, TALON enables off-the-shelf large language models to perform pattern-aware and semantically aligned forecasting across diverse scenarios. Extensive experiments on multiple benchmarks demonstrate that TALON achieves state-of-the-art accuracy while maintaining high efficiency and scalability. It also generalizes well in zero-shot settings and seamlessly incorporates both numerical and textual temporal cues. In future work, we plan to further improve pattern modeling via more adaptive and fine-grained mechanisms, and enhance domain transferability through efficient adaptation techniques.

## 7 ETHICS STATEMENT

This work focuses on adapting large language models to time series forecasting, with an emphasis on modeling temporal heterogeneity and semantic alignment. It relies solely on publicly available benchmark datasets that contain no personally identifiable or sensitive human data. No private or proprietary information was accessed or used. The study fully adheres to the ICLR Code of Ethics.

## 8 REPRODUCIBILITY STATEMENT

We provide detailed descriptions of the model architecture, training setup, and evaluation protocols in Sections 4 and 5. Hyperparameter settings, training configurations, and preprocessing pipelines are documented in Appendix A. Anonymized source code, configuration files, and reproduction scripts have be released at `https://anonymous.4open.science/r/TALON-BB00`. All benchmark datasets used in this work are publicly available. Furthermore, ablation studies and sensitivity analyses (Section 5.5 and Appendix D) demonstrate the robustness of our findings. These efforts collectively ensure that all reported results can be reliably reproduced.

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

## A    IMPLEMENTATION DETAILS

### A.1    BENCHMARK DATASETS

To evaluate the effectiveness and generalization ability of our proposed model, we conduct experiments on seven widely-used benchmark datasets, covering a diverse range of domains including electricity, traffic, and weather. The detailed dataset statistics are summarized in Table 7.

- **ETTh1 & ETTh2:** These datasets are part of the Electricity Transformer Temperature (ETT) benchmark, which records hourly temperature readings from two electricity transformers. Each dataset contains 7 variables.

- **ETTm1 & ETTm2:** These are the minute-level variants of the ETT benchmark, with a finer temporal granularity of 15 minutes. Each dataset contains 7 variables and significantly more samples due to the higher sampling rate.

- **Weather:** This dataset includes 21 meteorological variables, such as temperature, humidity, and wind speed, recorded every 10 minutes in 2020 at the Max Planck Biogeochemistry Institute's weather station.

- **Electricity:** This dataset records hourly electricity consumption for 321 clients. Due to its multivariate nature and high dimensionality, it is commonly used to evaluate model scalability and performance in high-dimensional forecasting tasks.

- **Traffic:** This dataset records hourly occupancy rates from 862 road sensors on freeways in the San Francisco Bay Area, spanning from January 2015 to December 2016. Its high dimensionality and complex temporal patterns make it a challenging benchmark for multivariate long-term forecasting.

We follow the same data processing and train-validation-test set split protocol used in TimesNet Wu et al. (2023), where the train, validation, and test datasets are strictly divided according to chronological order to ensure no data leakage. For long-term forecasting, we fix the context length of TALON and the lookback window of other baseline models to 672, while the prediction lengths vary among {96, 192, 336, 720}. Detailed settings are summarized in Table 7.

Table 7: Detailed dataset descriptions. Dim denotes the variate number. Dataset Size denotes the total number of time points in (Train, Validation, Test) split respectively. Forecast Length denotes the future time points to be predicted. Frequency denotes the sampling interval of time points.

| Dataset | Dim | Forecast Length | Dataset Size | Frequency | Information |
|---------|-----|-----------------|--------------|-----------|-------------|
| ETTh1 | 7 | {96, 192, 336, 720} | (8545, 2881, 2881) | 1 hour | Electricity |
| ETTh2 | 7 | {96, 192, 336, 720} | (8545, 2881, 2881) | 1 hour | Electricity |
| ETTm1 | 7 | {96, 192, 336, 720} | (34465, 11521, 11521) | 15 min | Electricity |
| ETTm2 | 7 | {96, 192, 336, 720} | (34465, 11521, 11521) | 15 min | Electricity |
| Weather | 21 | {96, 192, 336, 720} | (36792, 5271, 10540) | 10 min | Weather |
| Electricity | 321 | {96, 192, 336, 720} | (18317, 2633, 5261) | 1 hour | Electricity |
| Traffic | 862 | {96, 192, 336, 720} | (12185, 1757, 3509) | 1 hour | Transportation |

### A.2    IMPLEMENTATION DETAILS

TALON encodes statistical information in natural language form and uses a pretrained LLM (GPT2 Achiam et al. (2023)) to obtain prompt embeddings by extracting the final token's representation Liu et al. (2024d; 2025a). For multivariate forecasting, prompts are constructed independently for each variable and pre-tokenized to avoid runtime overhead.

After obtaining the prompt embeddings, TALON repurposes the LLM for time series forecasting. During training, only the parameters of the Heterogeneous Temporal Encoder and Forecast Head are updated, while the LLM remains frozen. At inference, TALON employs autoregressive decoding over language-aligned features to generate variable-length predictions without relying on textual prompts, ensuring efficient and scalable deployment.

All experiments are conducted using PyTorch Paszke et al. (2019) on NVIDIA A100 GPUs. We use the Adam optimizer Kingma & Ba (2014), with the initial learning rate randomly sampled from the range $[10^{-4}, 10^{-2}]$. Following the Channel Independence setting in Nie et al. (2023), each time series channel is modeled independently. The batch size is selected from $\{256, 384\}$, and each model is trained for 10 epochs. For evaluation, we rerun the baseline models using their official implementations. Specifically, most baselines are obtained from the TimesNet benchmark Wu et al. (2023) and the Timer_XL repository Liu et al. (2025e). For methods not included in these repositories, we follow the original official implementations released by the authors to ensure fair and consistent comparison.

## B  METRICS

**Mean Squared Error (MSE).** Mean Squared Error is one of the most widely used metrics for evaluating time series forecasting performance. It calculates the average of the squared differences between predicted values and ground truth values:

$$\text{MSE} = \frac{1}{N} \sum_{i=1}^{N} (y_i - \hat{y}_i)^2. \tag{12}$$

where $y_i$ and $\hat{y}_i$ denote the true and predicted values, respectively, and $N$ is the total number of predictions. MSE penalizes larger errors more severely, making it sensitive to outliers and suitable for applications that prioritize accurate modeling of extreme values.

**Mean Absolute Error (MAE).** Mean Absolute Error measures the average magnitude of the errors between predicted and true values, without considering their direction:

$$\text{MAE} = \frac{1}{N} \sum_{i=1}^{N} |y_i - \hat{y}_i|. \tag{13}$$

Compared to MSE, MAE is more robust to outliers and provides a direct interpretation of the average forecast error in the same units as the original data. It is especially useful when consistent accuracy across the entire forecast range is desired.

Both MSE and MAE are used in our evaluation to provide a comprehensive assessment of forecasting performance, balancing sensitivity to large deviations (MSE) and overall robustness (MAE).

**IMP.** IMP (Improvement) quantifies the relative performance gain of our proposed method (TALON) over each baseline method. Specifically, it denotes the average percentage reduction in both MSE and MAE across all seven datasets, defined as:

$$\text{IMP}_{\text{MSE}} = \frac{1}{D} \sum_{d=1}^{D} \frac{\text{MSE}_{\text{baseline}}^{(d)} - \text{MSE}_{\text{TALON}}^{(d)}}{\text{MSE}_{\text{baseline}}^{(d)}}, \tag{14}$$

$$\text{IMP}_{\text{MAE}} = \frac{1}{D} \sum_{d=1}^{D} \frac{\text{MAE}_{\text{baseline}}^{(d)} - \text{MAE}_{\text{TALON}}^{(d)}}{\text{MAE}_{\text{baseline}}^{(d)}}, \tag{15}$$

where $D$ is the number of datasets, and $\text{MSE}_{\text{baseline}}^{(d)}$ and $\text{MAE}_{\text{baseline}}^{(d)}$ refer to the error metrics of a given baseline on dataset $d$. Positive IMP values indicate that TALON achieves lower errors and thus better forecasting performance.

IMP provides a concise summary of overall improvement, enabling direct comparison of the relative effectiveness of TALON against each baseline across diverse datasets.

## C  TIME SERIES CHARACTERISTICS

We quantify the complexity of each univariate time series segment using three interpretable indicators: trend strength, local variation, and temporal dependency. Formally, for a univariate segment $s \in \mathbb{R}^S$, we extract the following:

**Trend Strength.** The trend of a time series refers to the long-term changes or patterns that occur over time. Intuitively, it represents the general direction in which the data is moving. Trend strength measures how much of the deseasonalized signal's variance can be explained by the underlying trend component. To compute it, we apply Seasonal-Trend decomposition using Loess (STL) to extract trend, seasonal, and residual components:

$$s = \text{Trend} + \text{Seasonal} + \text{Residual}. \tag{16}$$

We then calculate the deseasonalized signal $s' = s - \text{Seasonal}$ and define trend strength as:

$$\text{TrendStrength} = \max\left(0, 1 - \frac{\text{Var(Residual)}}{\text{Var}(s')}\right). \tag{17}$$

This formulation reflects the proportion of variance in the deseasonalized signal that is attributable to the trend component.

**Local Variation.** We compute the first-order difference $\Delta s_t = s_t - s_{t-1}$ and define local variation as:

$$\text{Variation} = \sigma(\text{lag}(1 + \text{std}(\Delta s)) - 1.0), \tag{18}$$

where $\sigma$ is the sigmoid function. This maps the log-scaled standard deviation to $[0,1]$ for robust normalization.

**Temporal Dependency.** We compute lag-1 autocorrelation:

$$\text{Autocorr} = |\text{acf}(s)[1]|, \tag{19}$$

where acf is the autocorrelation function. If the signal is constant or contains invalid values, the score is set to zero for robustness.

The final complexity descriptor is a 3-dimensional vector given by:

$$c = [c_1, c_2, c_3] \tag{20}$$
$$= [\text{TrendStrength}, \text{Variation}, \text{Autocorr}] \in [0,1]^3. \tag{21}$$

The full procedure for computing the statistical complexity descriptor is outlined in Algorithm 1.

---

**Algorithm 1** Statistical Complexity Computation for Time Series Patches

---

**Require:** A univariate time series patch $\mathbf{s} \in \mathbb{R}^S$
**Ensure:** A complexity vector $\mathbf{c} = [c_1, c_2, c_3] \in \mathbb{R}^3$
 1: **Trend Strength ($c_1$):**
 2: Apply STL decomposition on $\mathbf{s}$: $\mathbf{s} = \text{Trend} + \text{Seasonal} + \text{Residual}$
 3: Compute deseasonalized signal: $\mathbf{s}' = \mathbf{s} - \text{Seasonal}$
 4: **if** $\text{Var}(\mathbf{s}') = 0$ **then**
 5: $\quad c_1 \leftarrow 0$
 6: **else**
 7: $\quad c_1 \leftarrow 1 - \text{Var(Residual)}/\text{Var}(\mathbf{s}')$
 8: **end if**
 9: **Derivative Standard Deviation ($c_2$):**
10: Compute first-order difference: $\Delta \mathbf{s} = \mathbf{s}_{2:S} - \mathbf{s}_{1:S-1}$
11: $c_2 \leftarrow \log(1 + \text{std}(\Delta \mathbf{s}))$, then apply sigmoid scaling: $c_2 \leftarrow 1/(1 + \exp(-(c_2 - 1.0)))$
12: **Autocorrelation ($c_3$):**
13: Compute lag-1 autocorrelation:
14: $c_3 \leftarrow |\text{Corr}(\mathbf{s}_{1:S-1}, \mathbf{s}_{2:S})|$
15: **return** $\mathbf{c} = [c_1, c_2, c_3]$

---

# D SUPPLEMENTARY RESULTS

## D.1 TIME SERIES FORECASTING

We compare the performance of TALON with state-of-the-art LLM-based forecasting methods and well-acknowledged deep learning forecasters. Table 8 reports the results under the one-for-all forecasting setting across the ETT, ECL, Traffic, and Weather datasets. In this setup, each model is

Table 8: Multivariate forecasting (672-pred-96, 192, 336, 720) results under the one-for-all setting. Following Liu et al. (2024d), a single model is trained on a 96-step prediction horizon and evaluated on all horizons using rolling forecasting. The best results are in **bold**, and the second-best are *underlined*.

| Models | | LLM-based methods | | | | | | | | | | | | Deep learning forecasting methods | | | | | | | | | | | |
|---|---|---|---|---|---|---|---|---|---|---|---|---|---|---|---|---|---|---|---|---|---|---|---|---|---|
| | | TALON (Ours) | | LangTime (2025) | | CALF (2025b) | | AutoTimes (2024d) | | TimeLLM (2024a) | | FPT (2023) | | SimpleTM (2025) | | Timer_XL (2025e) | | TimeMixer (2024) | | iTransformer (2024c) | | PatchTST (2023) | | TimesNet (2023) | |
| Metric | | MSE | MAE | MSE | MAE | MSE | MAE | MSE | MAE | MSE | MAE | MSE | MAE | MSE | MAE | MSE | MAE | MSE | MAE | MSE | MAE | MSE | MAE | MSE | MAE |
| ETTh1 | 96 | **0.351** | **0.392** | 0.373 | 0.397 | 0.387 | 0.415 | 0.365 | 0.405 | 0.476 | 0.477 | 0.386 | 0.412 | 0.383 | 0.419 | 0.363 | 0.396 | 0.375 | 0.405 | 0.387 | 0.419 | 0.398 | 0.417 | 0.450 | 0.463 |
| | 192 | **0.381** | **0.415** | 0.404 | 0.416 | 0.397 | 0.421 | 0.396 | 0.423 | 0.545 | 0.517 | 0.422 | 0.433 | 0.416 | 0.439 | 0.404 | 0.423 | 0.409 | 0.426 | 0.421 | 0.440 | 0.432 | 0.441 | 0.471 | 0.475 |
| | 336 | **0.398** | **0.427** | 0.416 | 0.428 | 0.417 | 0.431 | 0.414 | 0.433 | 0.559 | 0.530 | 0.440 | 0.445 | 0.421 | 0.450 | 0.427 | 0.439 | 0.429 | 0.439 | 0.444 | 0.457 | 0.452 | 0.456 | 0.493 | 0.487 |
| | 720 | **0.414** | **0.446** | 0.433 | 0.447 | 0.462 | 0.450 | 0.432 | 0.452 | 0.588 | 0.558 | 0.440 | 0.460 | 0.477 | 0.491 | 0.436 | 0.458 | 0.458 | 0.466 | 0.474 | 0.490 | 0.483 | 0.492 | 0.567 | 0.532 |
| Avg. | | **0.386** | **0.420** | 0.406 | 0.422 | 0.416 | 0.429 | 0.402 | 0.428 | 0.542 | 0.520 | 0.422 | 0.437 | 0.424 | 0.450 | 0.407 | 0.429 | 0.418 | 0.434 | 0.432 | 0.451 | 0.441 | 0.451 | 0.495 | 0.489 |
| ETTh2 | 96 | 0.302 | 0.349 | 0.296 | 0.348 | 0.289 | **0.347** | **0.286** | 0.348 | 0.386 | 0.421 | 0.291 | 0.348 | 0.289 | 0.352 | 0.299 | 0.355 | 0.295 | 0.354 | 0.304 | 0.362 | 0.307 | 0.370 | 0.406 | 0.432 |
| | 192 | 0.355 | **0.388** | 0.370 | 0.397 | 0.376 | 0.400 | 0.371 | 0.408 | 0.404 | 0.435 | 0.368 | 0.399 | **0.353** | 0.399 | 0.367 | 0.401 | 0.369 | 0.402 | 0.384 | 0.410 | 0.392 | 0.423 | 0.459 | 0.458 |
| | 336 | **0.371** | **0.406** | 0.385 | 0.414 | 0.392 | 0.458 | 0.420 | 0.453 | 0.411 | 0.447 | 0.400 | 0.430 | 0.393 | 0.439 | 0.393 | 0.428 | 0.408 | 0.435 | 0.431 | 0.443 | 0.419 | 0.447 | 0.452 | 0.466 |
| | 720 | **0.393** | **0.435** | 0.404 | 0.436 | 0.433 | 0.474 | 0.521 | 0.516 | 0.463 | 0.479 | 0.419 | 0.452 | 0.434 | 0.467 | 0.448 | 0.472 | 0.468 | 0.479 | 0.478 | 0.479 | 0.452 | 0.477 | 0.502 | 0.496 |
| Avg. | | **0.355** | **0.395** | 0.364 | 0.399 | 0.373 | 0.419 | 0.400 | 0.431 | 0.416 | 0.446 | 0.370 | 0.407 | 0.367 | 0.414 | 0.377 | 0.414 | 0.385 | 0.417 | 0.399 | 0.423 | 0.392 | 0.429 | 0.455 | 0.463 |
| ETTm1 | 96 | **0.278** | **0.339** | 0.329 | 0.364 | 0.312 | 0.362 | 0.297 | 0.350 | 0.385 | 0.406 | 0.295 | 0.356 | 0.285 | 0.345 | 0.296 | 0.347 | 0.319 | 0.361 | 0.313 | 0.368 | 0.297 | 0.354 | 0.390 | 0.396 |
| | 192 | **0.324** | **0.367** | 0.378 | 0.393 | 0.328 | 0.375 | 0.344 | 0.377 | 0.490 | 0.471 | 0.338 | 0.384 | 0.339 | 0.370 | 0.349 | 0.378 | 0.375 | 0.392 | 0.351 | 0.391 | 0.340 | 0.381 | 0.463 | 0.426 |
| | 336 | **0.358** | **0.388** | 0.407 | 0.413 | 0.364 | 0.458 | 0.380 | 0.398 | 0.504 | 0.481 | 0.377 | 0.410 | 0.369 | 0.404 | 0.387 | 0.402 | 0.428 | 0.418 | 0.387 | 0.413 | 0.374 | 0.401 | 0.533 | 0.454 |
| | 720 | **0.418** | **0.424** | 0.476 | 0.452 | 0.464 | 0.472 | 0.433 | 0.431 | 0.529 | 0.495 | 0.452 | 0.455 | 0.438 | 0.425 | 0.453 | 0.441 | 0.523 | 0.464 | 0.456 | 0.450 | 0.431 | 0.433 | 0.636 | 0.493 |
| Avg. | | **0.345** | **0.380** | 0.398 | 0.405 | 0.367 | 0.417 | 0.364 | 0.389 | 0.477 | 0.463 | 0.365 | 0.401 | 0.358 | 0.386 | 0.371 | 0.392 | 0.411 | 0.409 | 0.377 | 0.405 | 0.360 | 0.392 | 0.505 | 0.442 |
| ETTm2 | 96 | **0.173** | **0.260** | 0.175 | 0.266 | 0.186 | 0.263 | 0.184 | 0.265 | 0.228 | 0.311 | 0.177 | 0.266 | 0.177 | 0.265 | 0.185 | 0.270 | 0.178 | 0.264 | 0.180 | 0.274 | 0.186 | 0.276 | 0.195 | 0.285 |
| | 192 | **0.223** | **0.299** | 0.228 | 0.301 | 0.268 | 0.327 | 0.247 | 0.307 | 0.271 | 0.338 | 0.244 | 0.310 | 0.237 | 0.306 | 0.247 | 0.312 | 0.242 | 0.306 | 0.240 | 0.312 | 0.247 | 0.318 | 0.253 | 0.323 |
| | 336 | **0.278** | **0.333** | 0.280 | 0.335 | 0.293 | 0.377 | 0.298 | 0.341 | 0.318 | 0.366 | 0.302 | 0.350 | 0.290 | 0.340 | 0.304 | 0.348 | 0.299 | 0.343 | 0.301 | 0.353 | 0.303 | 0.355 | 0.314 | 0.362 |
| | 720 | **0.362** | **0.383** | 0.365 | 0.391 | 0.376 | 0.395 | 0.378 | 0.395 | 0.422 | 0.420 | 0.410 | 0.423 | 0.369 | 0.391 | 0.389 | 0.402 | 0.391 | 0.405 | 0.407 | 0.416 | 0.397 | 0.414 | 0.411 | 0.420 |
| Avg. | | **0.259** | **0.319** | 0.262 | 0.323 | 0.281 | 0.341 | 0.277 | 0.327 | 0.310 | 0.359 | 0.283 | 0.337 | 0.268 | 0.325 | 0.281 | 0.333 | 0.277 | 0.330 | 0.282 | 0.338 | 0.284 | 0.341 | 0.293 | 0.347 |
| Weather | 96 | 0.161 | 0.213 | 0.168 | **0.207** | 0.168 | 0.221 | 0.166 | 0.221 | 0.208 | 0.263 | 0.169 | 0.230 | 0.169 | 0.217 | 0.286 | 0.334 | 0.168 | 0.214 | 0.172 | 0.224 | **0.159** | 0.214 | 0.169 | 0.228 |
| | 192 | **0.206** | **0.256** | 0.221 | 0.256 | 0.243 | 0.303 | 0.219 | 0.268 | 0.246 | 0.291 | 0.219 | 0.253 | 0.208 | 0.256 | 0.305 | 0.345 | 0.209 | 0.257 | 0.224 | 0.266 | 0.211 | 0.260 | 0.223 | 0.268 |
| | 336 | 0.258 | **0.296** | 0.284 | 0.302 | **0.256** | 0.315 | 0.277 | 0.311 | 0.286 | 0.319 | 0.268 | 0.305 | 0.265 | 0.306 | 0.330 | 0.358 | 0.261 | 0.298 | 0.283 | 0.305 | 0.268 | 0.303 | 0.288 | 0.308 |
| | 720 | **0.331** | **0.348** | 0.387 | 0.364 | 0.351 | 0.353 | 0.346 | 0.360 | 0.343 | 0.358 | 0.335 | 0.349 | 0.345 | 0.348 | 0.367 | 0.382 | 0.337 | 0.360 | 0.354 | 0.351 | 0.351 | 0.358 | 0.362 | 0.359 |
| Avg. | | **0.239** | **0.278** | 0.265 | 0.282 | 0.255 | 0.298 | 0.252 | 0.290 | 0.271 | 0.308 | 0.248 | 0.284 | 0.247 | 0.282 | 0.322 | 0.355 | 0.244 | 0.282 | 0.258 | 0.286 | 0.247 | 0.284 | 0.260 | 0.291 |
| Electricity | 96 | 0.133 | 0.227 | 0.144 | 0.240 | 0.133 | 0.230 | 0.135 | 0.230 | 0.139 | 0.243 | 0.138 | 0.237 | **0.131** | **0.226** | 0.137 | 0.230 | 0.136 | 0.227 | 0.135 | 0.231 | 0.136 | 0.240 | 0.182 | 0.287 |
| | 192 | **0.151** | **0.243** | 0.161 | 0.257 | 0.284 | 0.320 | 0.153 | 0.247 | 0.168 | 0.274 | 0.249 | 0.354 | 0.159 | 0.248 | 0.151 | 0.244 | 0.156 | 0.250 | 0.157 | 0.261 | | | 0.192 | 0.295 |
| | 336 | **0.163** | **0.260** | 0.180 | 0.275 | 0.276 | 0.316 | 0.172 | 0.266 | 0.184 | 0.283 | 0.280 | 0.381 | 0.172 | 0.268 | 0.169 | 0.274 | 0.170 | 0.260 | 0.172 | 0.267 | 0.182 | 0.288 | 0.201 | 0.303 |
| | 720 | **0.202** | **0.288** | 0.228 | 0.316 | 0.264 | 0.317 | 0.212 | 0.300 | 0.249 | 0.352 | 0.362 | 0.402 | 0.208 | 0.302 | 0.233 | 0.315 | 0.213 | 0.298 | 0.204 | 0.294 | 0.244 | 0.343 | 0.255 | 0.332 |
| Avg. | | **0.162** | **0.255** | 0.178 | 0.272 | 0.239 | 0.296 | 0.168 | 0.261 | 0.185 | 0.288 | 0.257 | 0.354 | 0.167 | 0.261 | 0.173 | 0.272 | 0.167 | 0.257 | 0.180 | 0.283 | 0.207 | 0.304 | | |
| Traffic | 96 | **0.338** | **0.232** | 0.379 | 0.254 | 0.355 | 0.249 | 0.347 | 0.249 | 0.383 | 0.264 | 0.384 | 0.278 | 0.410 | 0.306 | 0.347 | 0.245 | 0.418 | 0.311 | 0.350 | 0.257 | 0.374 | 0.273 | 0.602 | 0.317 |
| | 192 | **0.360** | 0.245 | 0.403 | 0.265 | 1.127 | 0.521 | 0.366 | 0.258 | 0.399 | 0.298 | 0.402 | 0.290 | 0.416 | 0.307 | 0.362 | **0.244** | 0.429 | 0.315 | 0.373 | 0.266 | 0.391 | 0.284 | 0.614 | 0.325 |
| | 336 | **0.374** | 0.249 | 0.424 | 0.275 | 1.136 | 0.522 | 0.383 | 0.267 | 0.423 | 0.323 | 0.427 | 0.311 | 0.437 | 0.318 | 0.381 | **0.255** | 0.442 | 0.321 | 0.390 | 0.274 | 0.409 | 0.299 | 0.618 | 0.329 |
| | 720 | **0.418** | 0.285 | 0.468 | 0.298 | 0.944 | 0.475 | 0.420 | 0.286 | 0.452 | 0.334 | 0.501 | 0.368 | 0.483 | 0.339 | 0.424 | **0.281** | 0.479 | 0.339 | 0.423 | 0.291 | 0.460 | 0.335 | 0.641 | 0.349 |
| Avg. | | **0.373** | **0.253** | 0.418 | 0.273 | 0.891 | 0.442 | 0.379 | 0.265 | 0.414 | 0.305 | 0.428 | 0.312 | 0.436 | 0.317 | 0.378 | 0.256 | 0.442 | 0.321 | 0.384 | 0.272 | 0.408 | 0.298 | 0.619 | 0.330 |

trained with a fixed input length of 672 and an output length of 96. During inference, we adopt a rolling forecasting strategy: the predicted values are iteratively appended to the input to reach the target forecast horizon.

In addition, we also evaluate the one-for-one setting, where separate models are trained for each forecast length. The corresponding results are provided in Table 9. All baselines are reproduced using their official implementations to ensure fair comparison.

## D.2 Zero-shot Forecasting

Following the zero-shot forecasting protocol proposed in AutoTimes Liu et al. (2024d), each experiment consists of a source dataset and a target dataset. The model is trained exclusively on the source dataset and directly applied to the target dataset without any fine-tuning or adaptation.

For the case of ETTh1 → ETTh2, the model is trained on ETTh1 and evaluated on ETTh2. We directly reuse the trained model from the one-for-all forecasting experiment reported in Table 8. The detailed results are presented in Table 10.

## D.3 Compared with MoE-based Methods

As shown in Table 11, TALON consistently outperforms four recent MoE-based methods across all datasets and prediction lengths. It achieves the lowest MSE in 13 out of 16 settings and ranks first in average MSE on every dataset. On average, TALON reduces the MSE by 10.7% compared to the baselines, demonstrating its strong modeling capability. This performance gain is attributed to the use of heterogeneous experts, which introduce diverse temporal inductive biases to better capture complex and non-stationary dynamics. Note that since the original TimeMoE paper does not report results trained on individual datasets, we adopt the TimeMoE results reported in MoFE-time, while other baselines use the results reported in their original papers.

Table 9: Multivariate forecasting (672-pred-{96, 192, 336, 720}) results under the one-for-one setting. A separate model is trained and evaluated for each prediction horizon. The best results are in **bold**, and the second-best are *underlined*.

| Models | | One-for-all | | Trained respectively on specific lookback / prediction length | | | | | | | | | | | | | | | | | | | | |
|---|---|---|---|---|---|---|---|---|---|---|---|---|---|---|---|---|---|---|---|---|---|---|---|---|
| | | TALON (Ours) | | LangTime (2025) | | CALF (2025b) | | AutoTimes (2024d) | | TimeLLM (2024a) | | FPT (2023) | | SimpleTM (2025) | | Timer_XL (2025e) | | TimeMixer (2024) | | iTransformer (2024c) | | PatchTST (2023) | | TimesNet (2023) | |
| Metric | | MSE | MAE | MSE | MAE | MSE | MAE | MSE | MAE | MSE | MAE | MSE | MAE | MSE | MAE | MSE | MAE | MSE | MAE | MSE | MAE | MSE | MAE | MSE | MAE |
| ETTh1 | 96 | **0.351** | **0.392** | 0.373 | 0.397 | 0.387 | 0.415 | 0.365 | 0.405 | 0.476 | 0.477 | 0.386 | 0.412 | 0.383 | 0.419 | 0.363 | 0.396 | 0.375 | 0.405 | 0.387 | 0.419 | 0.398 | 0.417 | 0.450 | 0.463 |
| | 192 | **0.381** | **0.415** | 0.402 | 0.427 | 0.415 | 0.433 | 0.456 | 0.469 | 0.596 | 0.533 | 0.425 | 0.435 | 0.409 | 0.434 | 0.425 | 0.438 | 0.410 | 0.433 | 0.422 | 0.443 | 0.441 | 0.450 | 0.468 | 0.476 |
| | 336 | **0.398** | **0.427** | 0.443 | 0.447 | 0.465 | 0.463 | 0.489 | 0.486 | 0.546 | 0.516 | 0.453 | 0.455 | 0.428 | 0.455 | 0.459 | 0.461 | 0.442 | 0.450 | 0.449 | 0.463 | 0.491 | 0.482 | 0.465 | 0.479 |
| | 720 | **0.414** | **0.446** | 0.588 | 0.517 | 0.494 | 0.498 | 0.516 | 0.506 | 0.692 | 0.589 | 0.486 | 0.483 | 0.470 | 0.489 | 0.554 | 0.524 | 0.483 | 0.482 | 0.547 | 0.534 | 0.540 | 0.520 | 0.553 | 0.540 |
| Avg. | | **0.386** | **0.420** | 0.451 | 0.447 | 0.440 | 0.452 | 0.457 | 0.466 | 0.578 | 0.529 | 0.438 | 0.446 | 0.422 | 0.449 | 0.450 | 0.455 | 0.428 | 0.442 | 0.451 | 0.465 | 0.468 | 0.467 | 0.484 | 0.489 |
| ETTh2 | 96 | 0.302 | 0.349 | 0.296 | 0.348 | 0.289 | 0.347 | 0.286 | 0.348 | 0.386 | 0.421 | 0.291 | 0.348 | 0.289 | 0.352 | 0.299 | 0.355 | 0.295 | 0.354 | 0.304 | 0.362 | 0.307 | 0.370 | 0.406 | 0.432 |
| | 192 | 0.355 | **0.388** | 0.396 | 0.404 | 0.355 | 0.388 | 0.387 | 0.414 | 0.426 | 0.446 | 0.378 | 0.418 | 0.354 | 0.392 | 0.358 | 0.395 | 0.368 | 0.398 | 0.380 | 0.408 | 0.416 | 0.432 | 0.444 | 0.462 |
| | 336 | 0.371 | 0.406 | 0.389 | 0.405 | 0.390 | 0.420 | 0.425 | 0.452 | 0.461 | 0.472 | 0.436 | 0.461 | 0.386 | 0.402 | 0.393 | 0.423 | 0.396 | 0.425 | 0.433 | 0.445 | 0.487 | 0.477 | 0.444 | 0.465 |
| | 720 | 0.393 | 0.435 | 0.471 | 0.473 | 0.430 | 0.455 | 0.460 | 0.470 | 0.465 | 0.482 | 0.479 | 0.482 | 0.416 | 0.434 | 0.429 | 0.457 | 0.436 | 0.460 | 0.482 | 0.488 | 0.460 | 0.474 | 0.439 | 0.460 |
| Avg. | | 0.355 | 0.395 | 0.388 | 0.408 | 0.366 | 0.402 | 0.390 | 0.421 | 0.435 | 0.455 | 0.396 | 0.427 | 0.361 | 0.395 | 0.370 | 0.407 | 0.374 | 0.409 | 0.400 | 0.426 | 0.417 | 0.438 | 0.433 | 0.455 |
| ETTm1 | 96 | **0.278** | **0.339** | 0.329 | 0.364 | 0.312 | 0.362 | 0.297 | 0.350 | 0.385 | 0.406 | 0.295 | 0.356 | 0.285 | 0.345 | 0.296 | 0.347 | 0.319 | 0.361 | 0.313 | 0.368 | 0.297 | 0.354 | 0.390 | 0.396 |
| | 192 | **0.324** | **0.367** | 0.398 | 0.399 | 0.343 | 0.382 | 0.396 | 0.413 | 0.392 | 0.410 | 0.332 | 0.378 | 0.346 | 0.388 | 0.332 | 0.376 | 0.373 | 0.398 | 0.347 | 0.386 | 0.389 | 0.404 | 0.527 | 0.467 |
| | 336 | **0.358** | **0.388** | 0.435 | 0.426 | 0.373 | 0.399 | 0.461 | 0.445 | 0.409 | 0.412 | 0.380 | 0.400 | 0.370 | 0.399 | 0.369 | 0.399 | 0.451 | 0.447 | 0.380 | 0.408 | 0.385 | 0.415 | 0.402 | 0.422 |
| | 720 | **0.418** | **0.424** | 0.498 | 0.466 | 0.424 | 0.427 | 0.490 | 0.464 | 0.438 | 0.439 | 0.429 | 0.425 | 0.424 | 0.428 | 0.441 | 0.441 | 0.529 | 0.488 | 0.448 | 0.450 | 0.477 | 0.463 | 0.456 | 0.450 |
| Avg. | | **0.345** | **0.380** | 0.415 | 0.414 | 0.363 | 0.393 | 0.411 | 0.418 | 0.406 | 0.417 | 0.359 | 0.390 | 0.356 | 0.390 | 0.359 | 0.391 | 0.418 | 0.423 | 0.372 | 0.403 | 0.387 | 0.409 | 0.444 | 0.434 |
| ETTm2 | 96 | **0.173** | **0.260** | 0.175 | 0.266 | 0.186 | 0.263 | 0.184 | 0.265 | 0.228 | 0.311 | 0.177 | 0.266 | 0.177 | 0.265 | 0.185 | 0.270 | 0.178 | 0.264 | 0.180 | 0.274 | 0.186 | 0.276 | 0.195 | 0.285 |
| | 192 | **0.223** | **0.299** | 0.243 | 0.312 | 0.236 | 0.299 | 0.285 | 0.338 | 0.253 | 0.320 | 0.243 | 0.309 | 0.237 | 0.310 | 0.230 | 0.302 | 0.235 | 0.301 | 0.243 | 0.316 | 0.246 | 0.312 | 0.281 | 0.339 |
| | 336 | **0.278** | **0.333** | 0.285 | 0.332 | 0.280 | 0.334 | 0.337 | 0.377 | 0.305 | 0.352 | 0.299 | 0.345 | 0.285 | 0.338 | 0.287 | 0.341 | 0.297 | 0.350 | 0.310 | 0.354 | 0.326 | 0.368 | 0.410 | 0.419 |
| | 720 | **0.362** | **0.383** | 0.362 | 0.384 | 0.364 | 0.389 | 0.423 | 0.433 | 0.373 | 0.398 | 0.375 | 0.401 | 0.364 | 0.390 | 0.406 | 0.407 | 0.376 | 0.401 | 0.375 | 0.401 | 0.414 | 0.432 | 0.410 | 0.419 |
| Avg. | | **0.259** | **0.319** | 0.266 | 0.323 | 0.266 | 0.321 | 0.307 | 0.353 | 0.290 | 0.345 | 0.274 | 0.330 | 0.269 | 0.329 | 0.276 | 0.329 | 0.269 | 0.327 | 0.274 | 0.335 | 0.289 | 0.343 | 0.303 | 0.353 |
| Weather | 96 | **0.161** | 0.213 | 0.168 | **0.207** | 0.168 | 0.221 | 0.166 | 0.221 | 0.208 | 0.263 | 0.169 | 0.230 | 0.169 | 0.217 | 0.286 | 0.334 | 0.168 | 0.214 | 0.172 | 0.224 | 0.159 | 0.214 | 0.169 | 0.228 |
| | 192 | **0.206** | **0.256** | 0.227 | 0.265 | 0.208 | 0.257 | 0.223 | 0.269 | 0.232 | 0.280 | 0.209 | 0.259 | 0.209 | 0.257 | 0.306 | 0.345 | 0.209 | 0.257 | 0.222 | 0.265 | 0.217 | 0.266 | 0.220 | 0.270 |
| | 336 | 0.258 | 0.296 | 0.326 | 0.331 | 0.253 | 0.295 | 0.264 | 0.302 | 0.259 | 0.325 | 0.256 | 0.299 | 0.263 | 0.300 | 0.332 | 0.361 | 0.249 | 0.289 | 0.288 | 0.309 | 0.249 | 0.290 | 0.265 | 0.302 |
| | 720 | **0.331** | 0.348 | 0.389 | 0.372 | 0.335 | 0.350 | 0.342 | 0.357 | 0.392 | 0.384 | 0.334 | 0.338 | 0.337 | 0.350 | 0.370 | 0.386 | 0.421 | 0.414 | 0.362 | 0.360 | 0.337 | 0.349 | 0.356 | 0.362 |
| Avg. | | **0.239** | **0.278** | 0.277 | 0.294 | 0.241 | 0.281 | 0.249 | 0.287 | 0.273 | 0.313 | 0.242 | 0.282 | 0.244 | 0.281 | 0.324 | 0.356 | 0.261 | 0.290 | 0.261 | 0.290 | 0.240 | 0.280 | 0.252 | 0.290 |
| Electricity | 96 | 0.133 | 0.227 | 0.144 | 0.240 | 0.133 | 0.230 | 0.135 | 0.230 | 0.139 | 0.243 | 0.138 | 0.237 | 0.131 | 0.226 | 0.137 | 0.230 | 0.136 | 0.227 | 0.135 | 0.231 | 0.136 | 0.240 | 0.182 | 0.287 |
| | 192 | **0.151** | **0.243** | 0.166 | 0.260 | 0.159 | 0.262 | 0.157 | 0.263 | 0.164 | 0.273 | 0.154 | 0.251 | 0.155 | 0.244 | 0.155 | 0.246 | 0.152 | 0.244 | 0.154 | 0.249 | 0.156 | 0.255 | 0.196 | 0.303 |
| | 336 | **0.163** | **0.260** | 0.175 | 0.272 | 0.174 | 0.270 | 0.176 | 0.275 | 0.179 | 0.284 | 0.167 | 0.265 | 0.166 | 0.264 | 0.168 | 0.266 | 0.169 | 0.264 | 0.169 | 0.266 | 0.170 | 0.273 | 0.204 | 0.308 |
| | 720 | 0.202 | 0.288 | 0.212 | 0.301 | 0.195 | 0.286 | 0.219 | 0.306 | 0.223 | 0.310 | 0.205 | 0.297 | 0.213 | 0.311 | 0.210 | 0.291 | 0.206 | 0.295 | 0.193 | 0.288 | 0.203 | 0.300 | 0.229 | 0.328 |
| Avg. | | **0.162** | **0.255** | 0.174 | 0.268 | 0.165 | 0.262 | 0.172 | 0.269 | 0.176 | 0.276 | 0.166 | 0.263 | 0.166 | 0.261 | 0.170 | 0.258 | 0.166 | 0.257 | 0.163 | 0.258 | 0.166 | 0.267 | 0.203 | 0.307 |
| Traffic | 96 | **0.338** | **0.232** | 0.379 | 0.254 | 0.355 | 0.249 | 0.347 | 0.249 | 0.383 | 0.264 | 0.384 | 0.278 | 0.410 | 0.306 | 0.347 | 0.245 | 0.418 | 0.311 | 0.350 | 0.257 | 0.374 | 0.273 | 0.602 | 0.317 |
| | 192 | **0.360** | **0.245** | 0.459 | 0.422 | 0.372 | 0.259 | 0.372 | 0.259 | 0.385 | 0.284 | 0.396 | 0.282 | 0.444 | 0.327 | 0.368 | 0.255 | 0.376 | 0.265 | 0.376 | 0.265 | 0.380 | 0.274 | 0.607 | 0.323 |
| | 336 | **0.374** | **0.249** | 0.507 | 0.402 | 0.389 | 0.267 | 0.389 | 0.271 | 0.403 | 0.284 | 0.407 | 0.286 | 0.468 | 0.344 | 0.386 | 0.264 | 0.390 | 0.275 | 0.390 | 0.272 | 0.402 | 0.288 | 0.633 | 0.333 |
| | 720 | **0.418** | **0.285** | 0.531 | 0.435 | 0.429 | 0.284 | 0.431 | 0.292 | 0.438 | 0.304 | 0.445 | 0.306 | 0.489 | 0.345 | 0.429 | 0.288 | 0.431 | 0.291 | 0.437 | 0.298 | 0.433 | 0.292 | 0.647 | 0.343 |
| Avg. | | **0.373** | **0.253** | 0.469 | 0.378 | 0.386 | 0.265 | 0.385 | 0.268 | 0.402 | 0.284 | 0.408 | 0.288 | 0.453 | 0.331 | 0.382 | 0.263 | 0.404 | 0.285 | 0.386 | 0.275 | 0.397 | 0.279 | 0.622 | 0.329 |

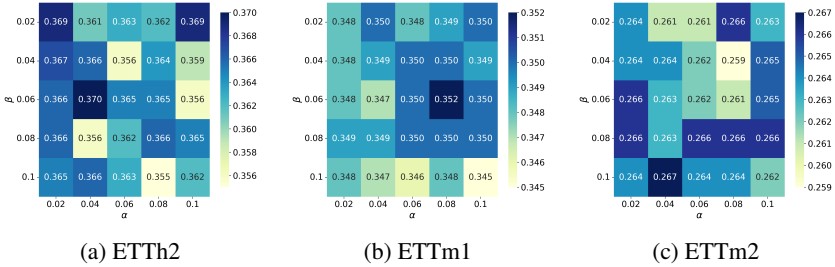

| (a) ETTh2 | (b) ETTm1 | (c) ETTm2 |
|---|---|---|

Figure 8: Parameter sensitivity of $\alpha$ and $\beta$ of the proposed method on the ETTh2, ETTm1, and ETTm2 datasets.

## D.4 GENERALITY

To evaluate the generality of TALON, we replace the default GPT-2 (124M) Achiam et al. (2023) backbone with several representative decoder-only LLMs: Qwen-0.5B Team (2024), Deepseek-1.5B Liu et al. (2024a), and LLaMA-7B Touvron et al. (2023).

We adopt AutoTimes as the baseline for comparison, as it is the strongest baseline in Table 8, where TALON achieves the smallest relative MSE reduction, making it a challenging reference point.

As shown in Table 12, TALON consistently outperforms AutoTimes across all datasets and prediction lengths, confirming that its design is broadly transferable and robust to the underlying language model. Interestingly, forecasting performance does not monotonically scale with model size: smaller models such as GPT-2 sometimes outperform larger ones like LLaMA-7B, suggesting that pretraining corpus, architectural choices, and tokenization strategies are critical factors beyond parameter count. While a systematic study of LLM scale is left for future work, our results demonstrate that TALON delivers consistent improvements across diverse backbones, highlighting its general applicability.

Table 10: Zero-shot forecasting result.

| Model | | | TALON (Ours) | | LangTime (2025) | | AutoTimes (2024d) | | Timer_XL (2025e) | |
|---|---|---|---|---|---|---|---|---|---|---|
| Metric | | | MSE | MAE | MSE | MAE | MSE | MAE | MSE | MAE |
| ETTh1 | ETTh2 | 96 | **0.290** | **0.348** | 0.338 | 0.373 | 0.294 | 0.352 | 0.305 | 0.358 |
| | | 192 | **0.350** | **0.387** | 0.418 | 0.417 | 0.354 | 0.388 | 0.372 | 0.400 |
| | | 336 | **0.378** | **0.411** | 0.429 | 0.427 | 0.383 | 0.416 | 0.397 | 0.425 |
| | | 720 | 0.410 | 0.439 | 0.430 | **0.435** | 0.409 | 0.439 | 0.420 | 0.450 |
| | Avg. | | **0.357** | **0.396** | 0.404 | 0.413 | 0.360 | 0.399 | 0.373 | 0.408 |
| | ETTm1 | 96 | **0.778** | 0.574 | 1.023 | 0.628 | 0.818 | **0.571** | 0.788 | 0.575 |
| | | 192 | **0.752** | 0.570 | 1.060 | 0.642 | 0.802 | **0.566** | 0.791 | 0.577 |
| | | 336 | **0.749** | **0.569** | 1.079 | 0.651 | 0.818 | 0.572 | 0.832 | 0.595 |
| | | 720 | **0.760** | **0.576** | 1.079 | 0.661 | 0.823 | 0.581 | 0.870 | 0.616 |
| | Avg. | | **0.760** | **0.572** | 1.060 | 0.646 | 0.815 | 0.572 | 0.820 | 0.591 |
| | ETTm2 | 96 | **0.226** | **0.316** | 0.290 | 0.354 | 0.242 | 0.327 | 0.240 | 0.324 |
| | | 192 | **0.281** | **0.349** | 0.358 | 0.390 | 0.307 | 0.363 | 0.304 | 0.362 |
| | | 336 | **0.335** | **0.380** | 0.422 | 0.422 | 0.368 | 0.397 | 0.367 | 0.398 |
| | | 720 | **0.427** | **0.429** | 0.543 | 0.478 | 0.456 | 0.444 | 0.463 | 0.449 |
| | Avg. | | **0.317** | **0.368** | 0.403 | 0.411 | 0.343 | 0.383 | 0.343 | 0.383 |
| ETTh2 | ETTh1 | 96 | 0.473 | 0.469 | 0.541 | 0.495 | 0.523 | 0.492 | **0.443** | **0.454** |
| | | 192 | 0.526 | **0.507** | 0.742 | 0.579 | 0.619 | 0.553 | 0.524 | 0.513 |
| | | 336 | **0.589** | **0.547** | 1.019 | 0.675 | 0.783 | 0.636 | 0.597 | 0.554 |
| | | 720 | **0.732** | **0.621** | 1.405 | 0.820 | 1.039 | 0.752 | 0.766 | 0.645 |
| | Avg. | | **0.580** | **0.536** | 0.927 | 0.642 | 0.741 | 0.608 | 0.583 | 0.542 |
| | ETTm1 | 96 | **0.703** | **0.538** | 1.052 | 0.608 | 1.250 | 0.662 | 0.782 | 0.578 |
| | | 192 | **0.739** | **0.564** | 0.996 | 0.605 | 1.055 | 0.634 | 0.831 | 0.610 |
| | | 336 | **0.791** | **0.594** | 0.956 | 0.605 | 0.972 | 0.639 | 0.864 | 0.636 |
| | | 720 | **0.854** | 0.630 | 0.952 | **0.624** | 0.991 | 0.687 | 0.931 | 0.676 |
| | Avg. | | **0.772** | **0.582** | 0.989 | 0.611 | 1.067 | 0.655 | 0.852 | 0.625 |
| | ETTm2 | 96 | **0.217** | **0.305** | 0.249 | 0.328 | 0.231 | 0.317 | 0.245 | 0.320 |
| | | 192 | **0.274** | **0.341** | 0.310 | 0.363 | 0.290 | 0.353 | 0.310 | 0.359 |
| | | 336 | **0.330** | **0.375** | 0.359 | 0.393 | 0.348 | 0.390 | 0.365 | 0.392 |
| | | 720 | **0.423** | **0.428** | 0.455 | 0.449 | 0.443 | 0.448 | 0.439 | 0.437 |
| | Avg. | | **0.311** | **0.362** | 0.343 | 0.383 | 0.328 | 0.377 | 0.340 | 0.377 |
| ETTm1 | ETTh1 | 96 | 0.638 | 0.552 | 0.619 | **0.515** | **0.597** | 0.522 | 0.718 | 0.586 |
| | | 192 | 0.623 | 0.541 | 0.649 | 0.526 | **0.608** | **0.525** | 0.712 | 0.584 |
| | | 336 | 0.618 | 0.541 | 0.644 | 0.527 | **0.607** | 0.529 | 0.739 | 0.598 |
| | | 720 | 0.616 | 0.550 | 0.637 | **0.535** | **0.609** | 0.547 | 0.764 | 0.617 |
| | Avg. | | 0.624 | 0.546 | 0.637 | **0.526** | **0.605** | **0.531** | 0.733 | 0.596 |
| | ETTh2 | 96 | **0.327** | **0.384** | 0.368 | 0.407 | 0.350 | 0.396 | 0.339 | 0.393 |
| | | 192 | **0.388** | **0.420** | 0.454 | 0.455 | 0.409 | 0.427 | 0.405 | 0.431 |
| | | 336 | **0.416** | **0.439** | 0.494 | 0.485 | 0.444 | 0.444 | 0.431 | 0.446 |
| | | 720 | **0.443** | **0.462** | 0.548 | 0.524 | 0.457 | 0.465 | 0.445 | 0.463 |
| | Avg. | | **0.393** | **0.426** | 0.466 | 0.468 | 0.412 | 0.433 | 0.405 | 0.433 |
| | ETTm2 | 96 | **0.187** | **0.273** | 0.215 | 0.294 | 0.192 | 0.275 | 0.203 | 0.284 |
| | | 192 | **0.247** | **0.311** | 0.285 | 0.338 | 0.258 | 0.315 | 0.269 | 0.325 |
| | | 336 | **0.300** | **0.344** | 0.343 | 0.375 | 0.314 | 0.349 | 0.325 | 0.358 |
| | | 720 | **0.383** | **0.394** | 0.431 | 0.426 | 0.396 | 0.399 | 0.406 | 0.408 |
| | Avg. | | **0.279** | **0.331** | 0.318 | 0.358 | 0.290 | 0.334 | 0.301 | 0.344 |
| ETTm2 | ETTh1 | 96 | **0.524** | **0.488** | 0.700 | 0.556 | 0.671 | 0.546 | 0.539 | 0.495 |
| | | 192 | **0.552** | **0.508** | 0.728 | 0.577 | 0.687 | 0.559 | 0.578 | 0.520 |
| | | 336 | **0.570** | **0.525** | 0.751 | 0.597 | 0.684 | 0.567 | 0.610 | 0.540 |
| | | 720 | **0.609** | **0.559** | 0.817 | 0.649 | 0.711 | 0.598 | 0.626 | 0.565 |
| | Avg. | | **0.563** | **0.520** | 0.749 | 0.595 | 0.688 | 0.568 | 0.588 | 0.530 |
| | ETTh2 | 96 | **0.285** | **0.350** | 0.336 | 0.385 | 0.315 | 0.375 | 0.286 | 0.353 |
| | | 192 | **0.352** | **0.392** | 0.402 | 0.421 | 0.370 | 0.409 | 0.357 | 0.398 |
| | | 336 | **0.387** | **0.416** | 0.430 | 0.444 | 0.394 | 0.427 | 0.411 | 0.431 |
| | | 720 | **0.402** | **0.437** | 0.475 | 0.477 | 0.440 | 0.462 | 0.416 | 0.450 |
| | Avg. | | **0.356** | **0.399** | 0.411 | 0.432 | 0.380 | 0.418 | 0.368 | 0.408 |
| | ETTm1 | 96 | **0.402** | **0.414** | 0.522 | 0.478 | 0.441 | 0.418 | 0.463 | 0.431 |
| | | 192 | **0.428** | **0.433** | 0.556 | 0.498 | 0.466 | 0.437 | 0.489 | 0.454 |
| | | 336 | **0.452** | **0.452** | 0.604 | 0.524 | 0.493 | 0.456 | 0.527 | 0.480 |
| | | 720 | **0.523** | **0.492** | 0.729 | 0.582 | 0.560 | 0.498 | 0.620 | 0.528 |
| | Avg. | | **0.452** | **0.448** | 0.603 | 0.520 | 0.490 | 0.452 | 0.525 | 0.473 |

## D.5 PARAMETER SENSITIVITY

**Sensitivity to $\alpha$ and $\beta$.** As shown in Figure 8, we further investigate the sensitivity of the hyper-parameters $\alpha$ and $\beta$ on three additional datasets: ETTh2, ETTm1, and ETTm2. Across all datasets, our method exhibits strong robustness to a wide range of $\alpha$ and $\beta$ values. The MSE variation across the grid is minimal (mostly within 0.01), indicating stable performance regardless of exact hyperparameter choices. Although slight differences exist in the optimal setting per dataset (e.g.,

Table 11: Comparison between TALON and MoE-based methods. The best results are in **bold**, and the second-best are *underlined*.

| Models | | TALON (Ours) | | FreqMoE (2025) | | MoFE-time (2025d) | | TimeMoE (2025) | | TFPS (2024) | |
|---|---|---|---|---|---|---|---|---|---|---|---|
| Metric | | MSE | MAE | MSE | MAE | MSE | MAE | MSE | MAE | MSE | MAE |
| ETTh1 | 96 | 0.351 | 0.392 | 0.371 | 0.388 | **0.337** | **0.380** | 0.360 | 0.396 | 0.398 | 0.413 |
| | 192 | **0.381** | 0.415 | 0.426 | 0.422 | 0.381 | **0.411** | 0.386 | 0.413 | 0.423 | 0.423 |
| | 336 | **0.398** | **0.427** | 0.475 | 0.447 | 0.414 | 0.436 | 0.407 | 0.433 | 0.484 | 0.461 |
| | 720 | **0.414** | **0.446** | 0.488 | 0.459 | 0.453 | 0.466 | 0.457 | 0.476 | 0.488 | 0.476 |
| Avg. | | **0.386** | **0.420** | 0.440 | 0.429 | 0.396 | 0.423 | 0.402 | 0.429 | 0.448 | 0.443 |
| ETTh2 | 96 | 0.302 | 0.349 | **0.287** | **0.337** | 0.307 | 0.352 | 0.352 | 0.388 | 0.313 | 0.355 |
| | 192 | **0.355** | 0.388 | 0.361 | **0.386** | 0.389 | 0.418 | 0.425 | 0.434 | 0.405 | 0.410 |
| | 336 | **0.371** | **0.406** | 0.407 | 0.423 | 0.514 | 0.480 | 0.526 | 0.485 | 0.392 | 0.415 |
| | 720 | **0.393** | 0.435 | 0.414 | 0.438 | 0.543 | 0.505 | 0.585 | 0.526 | 0.410 | **0.433** |
| Avg. | | **0.355** | 0.401 | 0.367 | **0.396** | 0.438 | 0.439 | 0.472 | 0.458 | 0.380 | 0.403 |
| ETTm1 | 96 | **0.278** | **0.339** | 0.314 | 0.356 | 0.294 | 0.352 | 0.319 | 0.373 | 0.327 | 0.367 |
| | 192 | **0.324** | **0.367** | 0.356 | 0.380 | 0.333 | 0.381 | 0.359 | 0.401 | 0.374 | 0.395 |
| | 336 | **0.358** | **0.388** | 0.385 | 0.404 | 0.400 | 0.433 | 0.404 | 0.433 | 0.401 | 0.408 |
| | 720 | **0.418** | **0.424** | 0.446 | 0.445 | 0.536 | 0.514 | 0.545 | 0.501 | 0.479 | 0.456 |
| Avg. | | **0.345** | **0.380** | 0.375 | 0.396 | 0.391 | 0.420 | 0.407 | 0.427 | 0.395 | 0.407 |
| ETTm2 | 96 | 0.173 | 0.260 | 0.173 | 0.266 | 0.189 | 0.278 | 0.258 | 0.320 | **0.170** | **0.255** |
| | 192 | **0.230** | 0.299 | 0.235 | 0.310 | 0.249 | 0.327 | 0.270 | 0.338 | 0.235 | **0.296** |
| | 336 | **0.282** | **0.333** | 0.290 | 0.350 | 0.294 | 0.356 | 0.365 | 0.405 | 0.297 | 0.335 |
| | 720 | **0.362** | **0.383** | 0.385 | 0.424 | 0.381 | 0.425 | 0.403 | 0.445 | 0.401 | 0.397 |
| Avg. | | **0.262** | **0.319** | 0.271 | 0.338 | 0.278 | 0.347 | 0.324 | 0.377 | 0.276 | 0.321 |

Table 12: Generality evaluation of TALON across different decoder-only LLM backbones on four benchmark datasets. TALON consistently improves upon the strong baseline AutoTimes across all settings, demonstrating robust transferability and model-agnostic behavior. The best results are in **bold**, and the second-best are *underlined*.

| Models | | AutoTimes | | GPT-2 (124M) | | Qwen-0.5B | | Deepseek-1.5B | | LLaMA-7B | |
|---|---|---|---|---|---|---|---|---|---|---|---|
| Metric | | MSE | MAE | MSE | MAE | MSE | MAE | MSE | MAE | MSE | MAE |
| ETTh1 | 96 | 0.365 | 0.405 | **0.351** | **0.392** | 0.360 | 0.396 | 0.362 | 0.399 | 0.358 | 0.394 |
| | 192 | 0.396 | 0.423 | **0.381** | 0.415 | 0.387 | **0.414** | 0.390 | 0.416 | 0.382 | 0.418 |
| | 336 | 0.414 | 0.433 | 0.398 | 0.427 | 0.403 | 0.425 | 0.403 | 0.425 | **0.393** | **0.421** |
| | 720 | 0.432 | 0.452 | 0.414 | 0.446 | **0.410** | **0.440** | 0.419 | 0.443 | 0.412 | 0.441 |
| Avg. | | 0.402 | 0.428 | **0.386** | 0.420 | 0.390 | **0.419** | 0.393 | 0.421 | 0.386 | 0.419 |
| ETTh2 | 96 | 0.286 | 0.348 | 0.302 | 0.349 | 0.286 | 0.348 | 0.283 | 0.347 | **0.282** | **0.346** |
| | 192 | 0.371 | 0.408 | 0.355 | 0.388 | 0.346 | 0.389 | **0.340** | **0.386** | 0.350 | 0.392 |
| | 336 | 0.420 | 0.453 | 0.371 | **0.406** | 0.370 | 0.412 | **0.361** | 0.406 | 0.376 | 0.418 |
| | 720 | 0.521 | 0.516 | **0.393** | **0.435** | 0.418 | 0.451 | 0.407 | 0.445 | 0.433 | 0.462 |
| Avg. | | 0.400 | 0.431 | 0.355 | **0.395** | 0.355 | 0.400 | **0.348** | 0.396 | 0.360 | 0.405 |
| ETTm1 | 96 | 0.297 | 0.350 | **0.278** | **0.339** | 0.289 | 0.345 | 0.291 | 0.348 | 0.285 | 0.345 |
| | 192 | 0.344 | 0.377 | **0.324** | **0.367** | 0.334 | 0.373 | 0.329 | 0.371 | 0.333 | 0.373 |
| | 336 | 0.380 | 0.398 | **0.358** | **0.388** | 0.368 | 0.394 | 0.362 | 0.391 | 0.371 | 0.395 |
| | 720 | 0.433 | 0.431 | 0.418 | 0.424 | 0.423 | 0.425 | **0.417** | **0.423** | 0.434 | 0.431 |
| Avg. | | 0.364 | 0.389 | **0.345** | **0.380** | 0.353 | 0.384 | 0.350 | 0.383 | 0.356 | 0.386 |
| ETTm2 | 96 | 0.184 | 0.265 | **0.173** | **0.260** | 0.177 | 0.262 | 0.177 | 0.266 | 0.174 | 0.263 |
| | 192 | 0.247 | 0.307 | **0.223** | **0.299** | 0.240 | 0.305 | 0.230 | 0.302 | 0.232 | 0.302 |
| | 336 | 0.298 | 0.341 | **0.278** | **0.333** | 0.296 | 0.342 | 0.283 | 0.335 | 0.283 | 0.335 |
| | 720 | 0.378 | 0.395 | **0.362** | **0.383** | 0.380 | 0.397 | 0.374 | 0.390 | 0.369 | 0.389 |
| Avg. | | 0.277 | 0.327 | **0.259** | **0.319** | 0.273 | 0.327 | 0.266 | 0.323 | 0.264 | 0.322 |

($\alpha = 0.06, \beta = 0.06$) on ETTm1), the overall insensitivity highlights that our method does not depend on meticulous tuning, making it practical and easy to deploy in real-world scenarios.

**Top-$k$ Expert Selection.** We conduct a sensitivity analysis on the top-$k$ parameter, which controls the number of activated experts during routing. As shown in Table 13, both $k = 2$ and $k = 3$ achieve

Table 13: Parameter sensitivity of $k$ of TALON on the ETTh1, ETTh2, ETTm1, and ETTm2 datsets.

|  | $H$ | $k=1$ MSE | $k=1$ MAE | $k=2$ MSE | $k=2$ MAE | $k=3$ MSE | $k=3$ MAE |
|---|---|---|---|---|---|---|---|
| ETTh1 | 96 | 0.358 | 0.397 | 0.360 | 0.397 | **0.351** | **0.392** |
|  | 192 | 0.389 | 0.419 | 0.391 | 0.418 | **0.381** | **0.415** |
|  | 336 | 0.409 | 0.433 | 0.409 | 0.431 | **0.398** | **0.427** |
|  | 720 | 0.433 | 0.455 | 0.429 | 0.449 | **0.414** | **0.446** |
| Avg. |  | 0.397 | 0.426 | 0.397 | 0.424 | **0.386** | **0.420** |
| ETTh2 | 96 | **0.282** | **0.346** | 0.302 | 0.349 | 0.293 | 0.352 |
|  | 192 | **0.345** | 0.390 | 0.355 | **0.388** | 0.353 | 0.397 |
|  | 336 | 0.374 | 0.418 | **0.371** | **0.406** | 0.379 | 0.423 |
|  | 720 | 0.432 | 0.462 | **0.393** | **0.435** | 0.434 | 0.465 |
| Avg. |  | 0.358 | 0.404 | **0.355** | **0.395** | 0.365 | 0.409 |
| ETTm1 | 96 | 0.282 | 0.342 | **0.278** | **0.339** | 0.278 | 0.340 |
|  | 192 | 0.325 | 0.369 | **0.324** | **0.367** | 0.328 | 0.370 |
|  | 336 | 0.365 | 0.392 | **0.358** | **0.388** | 0.364 | 0.391 |
|  | 720 | **0.413** | **0.421** | 0.418 | 0.424 | 0.424 | 0.425 |
| Avg. |  | 0.346 | 0.381 | **0.345** | **0.380** | 0.348 | 0.381 |
| ETTm2 | 96 | 0.194 | 0.282 | **0.172** | **0.259** | 0.173 | 0.260 |
|  | 192 | 0.262 | 0.326 | 0.230 | 0.299 | **0.223** | **0.299** |
|  | 336 | 0.333 | 0.369 | 0.283 | 0.334 | **0.278** | **0.333** |
|  | 720 | 0.438 | 0.429 | **0.361** | 0.385 | 0.362 | **0.383** |
| Avg. |  | 0.307 | 0.352 | 0.261 | 0.319 | **0.259** | **0.319** |
| $1^{st}$ Count |  | 3 | 2 | **9** | **9** | 8 | **9** |

competitive performance across most datasets and prediction lengths. Specifically, $k=2$ yields the most first-place results overall (9 for both MSE and MAE), while $k=3$ also performs strongly (8 for MSE and 9 for MAE). This suggests that leveraging multiple experts generally improves the model's ability to capture heterogeneous temporal patterns, compared to using a single expert ($k=1$). Moreover, the performance remains relatively stable across different $k$ values, demonstrating the robustness of the expert routing mechanism.

## E  SHOWCASES

To further illustrate the forecasting quality of TALON, we randomly select representative prediction examples on three datasets: ETTh1, ETTm1, and Weather, each with a forecast horizon of 192 time steps. We compare TALON against three strong baselines: Timer_XL Liu et al. (2025e), AutoTimes Liu et al. (2024d), and PatchTST Nie et al. (2023). As shown in Figure 9, TALON consistently generates predictions that better align with the ground truth, particularly in segments exhibiting nonstationarity, local fluctuations, or abrupt structural shifts.

These improvements stem from TALON's Heterogeneous Temporal Encoder, which employs a mixture of diverse architectural primitives to accommodate varying levels of temporal complexity. This design allows TALON to flexibly capture sharp transitions, smooth trends, and localized irregularities, avoiding the modeling bias introduced by homogeneous structures. In contrast, methods like PatchTST and AutoTimes often rely on fixed patch tokenization or prompt-based representations, which may be less robust when faced with irregular periodicities or regime shifts.

Furthermore, TALON's language-aligned temporal encoding leverages pretrained LLMs to extract semantic representations from natural language descriptions of statistical characteristics. These prompt embeddings serve as informative priors that enhance the model's understanding of temporal structure. By incorporating natural language priors, TALON gains a higher-level understanding of variable dependencies and temporal structures, which is especially beneficial in noisy or nonstationary environments where conventional models may overfit or underfit critical dynamics.

Overall, these qualitative results validate TALON's design philosophy of semantic-informed, pattern-aware forecasting, demonstrating its strong generalization ability across diverse datasets and dynamic regimes.

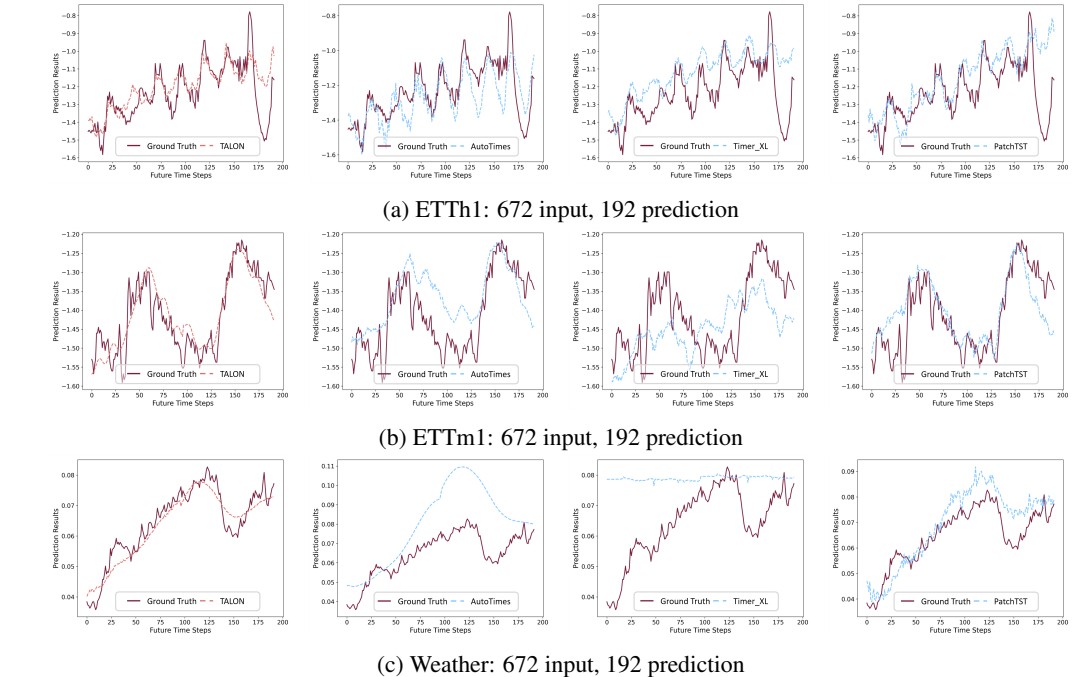

(a) ETTh1: 672 input, 192 prediction

(b) ETTm1: 672 input, 192 prediction

(c) Weather: 672 input, 192 prediction

Figure 9: Forecasting examples across ETTh1, ETTm1, and Weather datasets (672-step input, 192-step prediction).

# F BORADER IMPACT

## F.1 IMPACT ON REAL-WORLD APPLICATIONS

TALON's ability to align statistical time series features with natural language representations opens new avenues for integrating symbolic and numeric modalities in forecasting systems. This design makes it particularly suitable for real-world domains where both structured signals and contextual information (e.g., textual reports, user logs, or event annotations) coexist. For instance, in energy demand forecasting, TALON can incorporate external textual sources such as weather bulletins or maintenance notices, improving predictive accuracy during anomalous events. Similarly, in finance or supply chain domains, TALON offers a scalable and adaptable solution to model nonstationary dynamics without retraining for every configuration, thereby reducing operational cost and latency.

## F.2 IMPACT ON FUTURE RESEARCH

TALON bridges the gap between natural language processing and time series forecasting, contributing to the emerging paradigm of language-aligned modeling for structured signals. It introduces a flexible framework where natural language is not merely used as input, but also as a medium to encode domain knowledge in a human-interpretable way. This may inspire future work on hybrid modeling paradigms that combine statistical priors, expert annotations, and language reasoning for enhanced interpretability and adaptability. Additionally, the modular design of TALON, separating prompt encoding, temporal modeling, and autoregressive decoding, facilitates future integration with other modalities (e.g., vision or graphs), or with reinforcement learning for decision-aware forecasting.

# G THE USE OF LARGE LANGUAGE MODELS (LLMS)

Large Language Models (LLMs) were used in preparing this paper. Their role was limited to assisting with language polishing, such as improving grammar, refining phrasing, and enhancing readability of the manuscript. LLMs were not used for research ideation, methodological design, data

analysis, or experimental validation. All scientific content, ideas, and results are solely the work of the authors.

