# OpenReview forum: "Adapting LLMs to Time Series Forecasting via Temporal Heterogeneity Modeling and Semantic Alignment"
_ICLR.cc/2026/Conference — ICLR 2026 Conference Withdrawn Submission_

### Official Review · Reviewer_Vv7p · 2025-10-25

**Soundness:** 3
**Presentation:** 3
**Contribution:** 2
**Rating:** 4
**Confidence:** 5

**Summary:**

The paper proposes TALON, an LLM-based forecasting method, addressing temporal heterogeneity and the modality gap between continuous signals and discrete language. TALON introduces (1) a Heterogeneous Temporal Encoder that quantifies patch-level complexity and routes tokens to Linear/CNN/LSTM experts; (2) a Semantic Alignment Module that builds token-adaptive prompts from interpretable statistics and temporal context, then performs token-level contrastive alignment between time-series features and language embeddings; TALON has been evaluated across seven benchmarks under both one-for-all, one-for-one protocols, and zero-shot setting.

**Strengths:**

1. Prompt-free inference with token-level alignment: Avoids handcrafted prompts and reduces input redundancy while semantically grounding features via contrastive alignment.
2. The architecture HTE explicitly models heterogeneity with adaptive routing and complementary experts, supported by balanced-load regularization and strong ablations.
3. TALON has competitive accuracy with a compact head and frozen LLM backbone (~1.7M params; fast inference).

**Weaknesses:**

1. Some baseline results, such as TIME-LLM, are lower than those reported in the original paper, and TALON shows no clear advantage compared to the original results. Certain related works or baselines, such as TEMPO [1] and [2], are also missing.
2. Token-adaptive prompts are not used during inference; the paper could further explore whether reintroducing lightweight textual cues would help in out-of-distribution settings. Context(relevant to time series)-aided time series forecasting would make the use of LLMs more motivated.
3. The ablations are insufficient, consider replacing the LLM component with simpler architectures to better demonstrate the necessity of using LLMs.


[1] TEMPO: Prompt-based Generative Pre-trained Transformer for Time Series Forecasting.

[2] From News to Forecast: Integrating Event Analysis in LLM-Based Time Series Forecasting with Reflection

**Questions:**

Have you considered using text related to time series to assist with time series forecasting? Some related works have provided such textual and time series data [1] [2] [3].

[1] Time-MMD: Multi-Domain Multimodal Dataset for Time Series Analysis

[2] From News to Forecast: Integrating Event Analysis in LLM-Based Time Series Forecasting with Reflection

[3] Intervention-Aware Forecasting: Breaking Historical Limits from a System Perspective

---

### Official Review · Reviewer_rsLS · 2025-10-26

**Soundness:** 3
**Presentation:** 3
**Contribution:** 2
**Rating:** 4
**Confidence:** 4

**Summary:**

This paper proposes TALON, aimed at adapting large-scale language models for time series forecasting tasks. The paper addresses two challenges: temporal heterogeneity in time series and the modal disparity between continuous signals and discrete language. TALON comprises three main modules: the Heterogeneous Time Encoder for local pattern modeling, the Semantic Alignment Module that embeds temporal features into linguistic semantic space via contrastive learning, and finally, an autoregressive prediction layer achieved through a frozen LLM and a Lightweight Forecasting Head. Systematic experiments across seven public benchmark datasets demonstrate that TALON achieves an average MSE improvement of approximately 10–20% over existing SOTA methods. It also exhibits strong performance in zero-shot transfer learning, efficiency, and ablation studies. Overall, the paper aligns with current research frontiers integrating LLMs with temporal data, featuring a framework with notable innovation and comprehensive experimental coverage.

**Strengths:**

1. Clear research motivation and structural design: The paper clearly articulates the structural and modal differences between time series tasks and language modeling, systematically addressing this issue through the HTE and SAM modules. The framework exhibits logical coherence with focused innovations.

2. Rational and interpretable module design: The HTE module employs local statistical features for expert dynamic routing, integrating linear, CNN, and LSTM experts to capture multi-scale temporal patterns. The SAM module introduces token-level alignment to enhance semantic consistency between temporal features and linguistic representations.

3. Comprehensive experimental coverage: The authors validate the model across seven standard datasets—including the ETT series, Weather, Electricity, and Traffic—and incorporate three prediction settings: one-for-all, one-for-one, and zero-shot. This demonstrates TALON's potential for generalization and stability.

**Weaknesses:**

1. Lack of Innovation and Differentiation: Although TALON conceptually integrates heterogeneous modeling with semantic alignment, its implementation primarily relies on existing MoE routing mechanisms and contrastive learning frameworks. The three features—trend, fluctuation, and autocorrelation—in the HTE module are relatively conventional, lacking exploration of more complex patterns such as frequency domain and long-range dependencies. We recommend the authors further elaborate on the theoretical rationale for these feature selections and validate their validity through feature importance or replacement experiments.

2. Shallow Semantic Alignment Mechanism: The SAM module performs cosine similarity alignment solely at the lexical token level, disregarding temporal context or cross-segment structural relationships. This alignment process fails to capture cross-layer feature consistency or global temporal constraints, undermining its claim of “semantically consistent prediction.” Consequently, the context modeling capabilities of pre-trained LLMs may not be directly applicable to modeling temporal relationships.

3. Semantic alignment may lead to excessive assimilation of temporal information: The paper's alignment mechanism minimizes the angular distance between temporal representations and prompt representations at the token level, then directly drives predictions using the “aligned representations” to drive the frozen LLM + linear head during inference. This introduces a structural risk: when the alignment loss accounts for a high proportion or training is imbalanced, the temporal branch representation may become overly close to the prompt representation. Consequently, independent temporal structure and statistical features may be lost, and the model may primarily rely on “semantic prompts” for prediction rather than utilizing pre-aligned temporal information. The current paper does not provide diagnostics or controls to rule out such “semantics-dominated, temporally weakened” scenarios.

4. Insufficient justification for LLM freezing strategy: Authors fully freeze the LLM, training only the encoder and prediction head, without analyzing whether lighter approaches like adapter/prefix tuning were attempted. If LLMs cannot dynamically adapt to numerical patterns, the conclusion that “LLMs can be directly applied to time series inference” appears overly optimistic. A parameter efficiency comparison is recommended to clarify the benefits and trade-offs of the freezing strategy.

**Questions:**

1. On Innovation and Feature Selection in HTE: Could the authors further explain the rationale behind choosing trend, fluctuation, and autocorrelation as the three primary descriptors in the HTE module? Have alternative or learned features—such as frequency-domain representations, spectral energy, or Transformer-based contextual embeddings—been evaluated for modeling complex or long-range dependencies? Additionally, would the authors consider adding a feature importance analysis or feature replacement study to validate the necessity of these hand-crafted statistical indicators?

2. On the Scope of Semantic Alignment in SAM: The SAM module aligns representations through token-level cosine similarity, which may overlook cross-temporal dependencies or higher-order contextual relationships. Could the authors clarify whether they attempted any context-aware alignment strategies? Furthermore, how do they ensure that global temporal consistency is maintained when alignment is only enforced locally at the token level?

3. On Potential Information Assimilation during Alignment: The contrastive alignment loss may drive the temporal encoder representations to become overly similar to semantic prompt embeddings, potentially leading to the loss of temporal structure. Have the authors conducted diagnostic experiments to confirm that the model still leverages temporal information after alignment—for example, by testing performance when prompts are shuffled or masked, or by measuring mutual information or representational similarity before and after alignment? Could they discuss mechanisms or regularization strategies that mitigate over-assimilation?

4. On the LLM Freezing Strategy and Adaptation Capability: The paper reports that the LLM backbone is fully frozen during training. Could the authors elaborate on whether lighter adaptation methods—such as adapter tuning, prefix tuning, or LoRA—were explored? How does freezing the LLM affect its ability to model numerical patterns, and have the authors compared parameter efficiency or inference latency under different tuning strategies? Providing such comparisons would clarify the trade-offs between efficiency and adaptability.

---

### Official Review · Reviewer_G2zG · 2025-10-29

**Soundness:** 3
**Presentation:** 3
**Contribution:** 2
**Rating:** 4
**Confidence:** 5

**Summary:**

This paper introduces TALON (Temporal-heterogeneity and Language-Oriented Network)—a new architecture that bridges large language models (LLMs) and time series forecasting! It tackles two fundamental challenges of temporal data: pattern heterogeneity and the modality gap between continuous signals and discrete language representations. TALON combines a Heterogeneous Temporal Encoder (HTE) for adaptive pattern extraction, a Semantic Alignment Module (SAM) for embedding-level cross-modal alignment, and a Lightweight Forecasting Head (LFH) that performs autoregressive prediction through a frozen LLM backbone.The paper contributes to the growing body of work exploring foundation models for structured data, providing clear architectural design, solid empirical validation, and insightful analyses of expert routing and cross-modal representation behavior! This research advances the understanding of how language priors can be leveraged to strengthen forecasting under distributional shifts and heterogeneous temporal regimes!

**Strengths:**

1. Comprehensive Temporal Modeling: By combining heterogeneous experts in the HTE module, TALON effectively captures diverse temporal behaviors including trends, local fluctuations, and long-range dependencies. This multi-expert design enhances its ability to adapt to nonstationary and complex temporal patterns.

2. Semantic–Numerical Integration: The Semantic Alignment Module (SAM) bridges numerical time-series representations with linguistic semantics through contrastive learning. This integration allows the model to leverage pretrained LLMs for high-level reasoning while preserving low-level temporal structure, improving interpretability and generalization.

3. Empirical Validation Across Benchmarks: Extensive experiments on seven benchmark datasets demonstrate consistent improvements (approximately 10–20% MSE reduction) over strong baselines in both multi-horizon and zero-shot forecasting settings. These results verify TALON’s robustness and transferability across domains and temporal resolutions.

**Weaknesses:**

1. Information Loss in Cross-Modal Alignment: During cross-modal contrastive alignment, temporal embeddings may suffer from information loss as the optimization drives them excessively close to the semantic embeddings. Consequently, the model’s final representations could become dominated by textual prompts, diminishing the independent contribution of the HTE module and weakening the preservation of intrinsic temporal dynamics. The paper lacks diagnostic experiments or regularization strategies that demonstrate the balance between alignment strength and temporal integrity.

2. Lack of Consistency Verification in Expert Routing: The HTE module determines expert selection using only three local statistical indicators—trend intensity, local fluctuation, and autocorrelation—but the paper does not examine whether this routing remains consistent under minor perturbations. For example, when applying slight noise to similar time series segments, it is unclear whether the same experts would be selected. Without such consistency verification, the robustness and interpretability of the heterogeneous expert routing mechanism remain uncertain.

3. Unanalyzed Contribution of Prompt Components: In the adaptive prompt construction process, the three components—routing prompts, temporal context, and complexity statistics—are combined without assessing their respective contributions. The absence of ablation or attribution analysis makes it difficult to determine which component primarily facilitates cross-modal alignment and whether textual or numerical features dominate semantic influence.

**Questions:**

1. On Temporal–Semantic Information Balance: How do the authors ensure that temporal embeddings retain sufficient information during cross-modal alignment? Have they measured representation diversity or mutual information before and after alignment, or applied constraints to prevent the temporal encoder from over-assimilating to the semantic space?

2. On Expert Routing Robustness: Have the authors evaluated whether HTE’s expert selection remains consistent for highly similar or noise-perturbed sequences? Could they provide quantitative or visual analyses verifying that routing decisions are stable and interpretable under small data perturbations?

3. On Component-wise Prompt Effectiveness: sCould the authors conduct ablation or attribution studies to quantify the relative contribution of routing prompts, temporal context, and statistical descriptors in the adaptive prompt design? This would clarify how textual versus numerical information contributes to cross-modal alignment and prediction accuracy.

---

### Official Review · Reviewer_QPnX · 2025-10-31

**Soundness:** 3
**Presentation:** 2
**Contribution:** 2
**Rating:** 4
**Confidence:** 4

**Summary:**

This paper proposes a novel framework called TALON (Temporal-heterogeneity And Language-Oriented Network), which addresses the challenges of temporal heterogeneity and the modality gap when applying large language models (LLMs) to time series forecasting tasks. The paper demonstrates the effectiveness of TALON through baseline comparison experiments, validates the contribution of each component via ablation studies, and shows that TALON maintains high efficiency with minimal overhead during the inference stage.

**Strengths:**

1. I agree that there is currently a significant gap between LLMs and time series forecasting. Existing LLM-based methods—including reprogramming approaches such as Time-LLM and prompt-based approaches such as LLMTime—indeed have certain limitations. The paper identifies two reasonable underlying causes, namely temporal heterogeneity and modality differences, which are accurately and clearly articulated in the introduction.
2. The proposed method adopts a training–inference decoupled design, functioning as a plug-and-play module placed in front of a frozen LLM. It supports one-time training that generalizes across all time series benchmarks, and the experiments validate its effectiveness. This design is generalizable and computationally efficient, reducing the overall resource cost.
3. It is interesting to see this work uses contrastive learning to align time series representations with text representations instead of brutally train the LLM on time series data and hoping it can automatically learn the new tokens.

**Weaknesses:**

1. (major) Choice of mode quantization features: The MoE-like architectural design of this paper is impressive. However, the selection of time series statistical features—trend strength, local variation, and autocorrelation—appears somewhat simplistic. These statistics may not be sufficient to uniquely or accurately characterize the properties of a time series patch. It remains unclear whether incorporating additional or more informative statistical features could provide a more precise representation of patches and potentially improve the final prediction performance.
2. (major) It is still unclear that if the contrastive trained time series token can be directly applied in the next-token-prediction-based LLMs. Further, since the time series tokens are only corasely aligned with some statistical features, the granularity of its semantic should be very coarse and it is not sure if these tokens still hold their previous time series details.
3. (major) Concern regarding ablation studies and hyperparameter sensitivity analysis:

    In the ablation study, the authors show that removing individual modules or any one of the three experts (Linear, CNN, LSTM) leads to a decline in predictive performance. However, the degree of degradation is very small—on the order of 0.01 in MSE and 0.001 in MAE. Such minimal performance differences somewhat weaken the justification for the model’s highly complex architectural design.

    Furthermore, The loss function combines multiple components and includes two manually defined hyperparameters, α and β, which are not learnable. Although the authors perform a sensitivity analysis, but as shown in Figure 7, the MSE exceeds 0.390 under most combinations of α and β, and the magnitude of this variation is even larger than that observed in some groups of the ablation study (see Tables 5 and 6). Hence, the results do not convincingly support the claim that “TALON is robust to α and β.”

    If the authors consider such fluctuations in acceptable in the sensitivity analysis, it would further weaken the argument for the necessity of the model’s complex modules, since their removal only slightly affects performance within tolerable bounds. If the authors consider an MSE change of 0.01 to be meaningful in the ablation study, then this level of variation would cast doubt on the conclusion that “TALON is robust to α and β.”

4. (minor) Lack of analysis on training cost:

    While the paper discusses the efficiency of TALON during inference, it does not analyze the training cost in depth. The SAM module’s patch-wise dynamic prompt generation and contrastive learning introduce substantial computational overhead. The authors should include metrics such as training time and GPU memory consumption to provide a more comprehensive evaluation of TALON’s efficiency.

**Questions:**

See the weakness.

---

### Note · Authors · 2025-12-28

I have read and agree with the venue's withdrawal policy on behalf of myself and my co-authors.